ecology/microbiology

bacteria, coral reefs, microbiomes, microbial ecology, phycobiome

**Author for correspondence:**
Amy A. Briggs
e-mail: amy.briggs@uga.edu

# Local versus site-level effects of algae on coral microbial communities

Amy A. Briggs[1], Anya L. Brown[1,2,3] and Craig W. Osenberg[1]

[1]Odum School of Ecology, University of Georgia, Athens, GA, USA
[2]Woods Hole Oceanographic Institution, Woods Hole, MA, USA
[3]School of Natural Resources and Environment, University of Florida, USA

AAB, 0000-0002-5782-7928; ALB, 0000-0002-0436-1458;
CWO, 0000-0003-1918-7904

Microbes influence ecological processes, including the dynamics and health of macro-organisms and their interactions with other species. In coral reefs, microbes mediate negative effects of algae on corals when corals are in contact with algae. However, it is unknown whether these effects extend to larger spatial scales, such as at sites with high algal densities. We investigated how local algal contact and site-level macroalgal cover influenced coral microbial communities in a field study at two islands in French Polynesia, Mo'orea and Mangareva. At 5 sites at each island, we sampled prokaryotic microbial communities (microbiomes) associated with corals, macroalgae, turf algae and water, with coral samples taken from individuals that were isolated from or in contact with turf or macroalgae. Algal contact and macroalgal cover had antagonistic effects on coral microbiome alpha and beta diversity. Additionally, coral microbiomes shifted and became more similar to macroalgal microbiomes at sites with high macroalgal cover and with algal contact, although the microbial taxa that changed varied by island. Our results indicate that coral microbiomes can be affected by algae outside of the coral's immediate vicinity, and local- and site-level effects of algae can obscure each other's effects when both scales are not considered.

## 1. Introduction

Coral reefs around the world are increasingly faced with disturbances such as bleaching events, hurricanes, disease outbreaks, overfishing and eutrophication, which can permit the rapid proliferation of fleshy algae [1–6]. These increases in algae have important consequences for corals, which compete with algae for space, but are much more slow-growing, and thus take longer to recover when their populations decline. Besides pre-empting space on reefs,

algae have a variety of other negative effects on corals, including reducing coral recruitment, growth and survival, often mediated through direct mechanisms such as overgrowth, abrasion and allelopathy [7–10]. However, indirect effects of algae mediated through microbes are increasingly recognized as an important avenue through which algae can harm corals and potentially reduce their abundance [11,12].

Corals typically have consistent associations with certain microbial taxa [13–15], many of which influence coral physiology, including internal nutrient cycling, production of enzymes and essential vitamins, as well as resistance to pathogen invasion (reviewed in [16]). Algae can influence the microbial communities associated with corals, i.e. their microbiomes [17–19]. This effect on coral microbiomes could occur because algae leak dissolved organic carbon (DOC), allelochemicals, and other secondary metabolites that can alter the growth and composition of microbes on the surface of the algae and in the surrounding seawater [10,20,21], and these changes may subsequently affect the coral microbiome. In particular, algae often facilitate the growth of copiotrophs, fast-growing heterotrophic taxa that do best under high nutrient environments, as well as potential pathogens [10,22,23]. Since alterations to the coral microbiome can affect the coral's sensitivity to abiotic stressors [24–26], its susceptibility to disease [12,27–29], and its physiological performance and survival [30], microbiome shifts associated with algae could have significant consequences for corals.

As algae increase in abundance on many reefs around the world, it is important to understand the spatial scale at which algae affect coral. However, the observational and experimental studies that have demonstrated effects of algae on coral microbiomes all focus on corals that were in close proximity to (e.g. <1 m) or in contact with algae: i.e. a 'local' effect [17–19]. Although there are no studies that document effects of algae on corals at larger 'regional' scales (e.g. at sites that are each 1000–10 000 m$^2$), Haas *et al*. [22] found that water column DOC, microbial density and microbial community composition were correlated with the regional per cent cover of algae (a proxy for regional algal biomass or density). Similarly, Kelly *et al*. [31] found shifts in near-benthos water microbial communities with changes in benthic community composition (in 20 m$^2$ areas). If these larger-scale effects of algae or benthic communities on water microbial communities also manifest on corals, then algae may alter coral microbiomes at both local and regional scales. However, dilution and benthic boundary layers might prevent effects at larger (i.e. regional) spatial scales. Furthermore, studies of corals in contact with algae suggest that changes in coral microbiomes disappear at distances greater than 5 cm from the zone of interaction between the algae and coral [32–34]. Unfortunately, no studies have directly evaluated if coral microbial communities change in response to regional algal density, or if the possible effects of algae at local and regional scales interact with one another.

Given current knowledge, there are several plausible hypotheses about the effects of local and regional algae on coral microbiomes. The first hypothesis is that only local algae have an effect and that there is no additional effect of regional algal density on the coral microbiome (figure 1*a*), possibly because of dilution or boundary layers, which would limit contact of benthic corals with microbes in the overlying waters. If, on the other hand, algae also have effects that increase with the regional density of algae, these regional and local effects could combine additively (figure 1*b*). Alternatively, local and regional effects could interact with one another (figure 1*c,d*). A synergistic interaction would occur if regional effects of algae are more pronounced when a coral is also in contact with algae (figure 1*c*). By contrast, an antagonistic interaction would occur if regional effects of algae are most pronounced in the absence of algal contact (figure 1*d*). Such antagonisms could arise if the coral microbiome response is saturated by the local effects of algae, causing an absence of any regional algal effect in the presence of algal contact. As a result, the response of corals not in contact with algae would converge on the response of corals experiencing algal contact as the regional abundance of algae increases (figure 1*d*). Of course, more complex interactions also are possible.

Effects of algae on the coral microbiome could also be assessed by comparing the coral microbiome of a coral exposed to algae to the pristine state of the coral (not exposed to algae) and the microbial assemblage of the algae. Exposure of a coral to algae might cause the coral microbiome to become intermediate in composition between that found on isolated corals versus on algae (electronic supplementary material, figure S1a). Alternatively, exposure to algae could act as a stressor that facilitates the development of a new microbiome, distinct from the ones typically found on either corals or algae (a deterministic shift; electronic supplementary material, figure S1b). A third possibility is that exposure to algae could disrupt the coral microbiome and allow idiosyncratic shifts in its composition that vary widely among similarly exposed corals (a stochastic shift; electronic supplementary material, figure S1c).

To determine whether algae affect coral microbiomes at multiple spatial scales and if coral microbiomes change in predictable ways in response to algae, we conducted a field study at two islands in French Polynesia, Mo'orea and Mangareva. At each island, we surveyed benthic

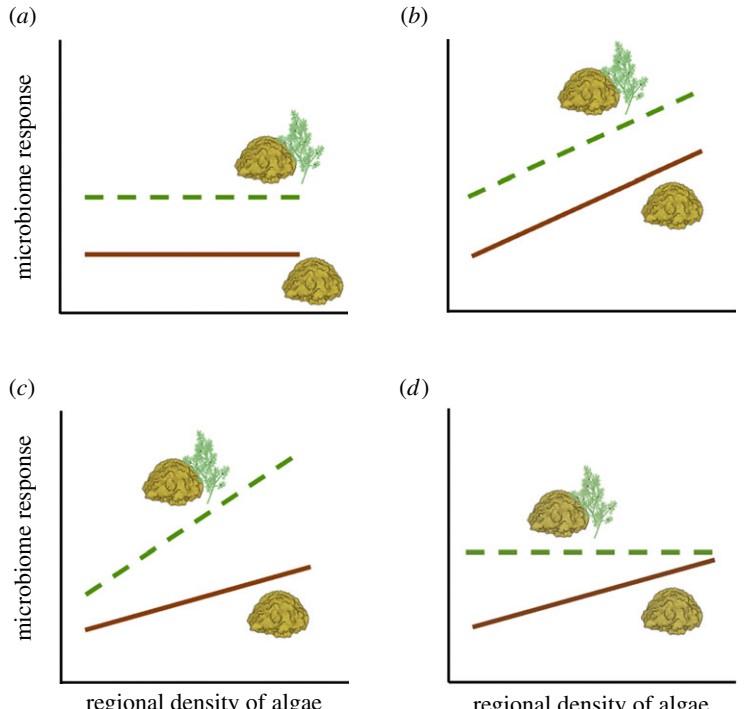

**Figure 1.** Hypothetical responses of coral microbiomes to algae at local and regional scales. Local effects are indicated by differences between the dashed green line (contact with algae) versus the solid brown line (no contact). Regional effects are depicted by the slopes of these lines: i.e. the response of the microbiome to increasing regional density of algae. (*a*) Local effects of algae without any regional effects; (*b*) additive local and regional effects; (*c*) synergistic local and regional effects; and (*d*) antagonistic local and regional effects.

communities and collected microbial samples at five sites that fell along a gradient of macroalgal cover (5–51%; electronic supplementary material, figure S2). We collected microbial samples from individual coral heads of the dominant coral species at each island, stratifying our sampling across corals that were either in contact with algae (i.e. touching) or not in contact with algae (i.e. at least 20 cm from the nearest alga). Because algal functional groups leak different types and amounts of chemical compounds [10,35], and typically have different biomasses on reefs, which could modify their effects on microbes, we sampled corals in contact with turf algae and corals in contact with the dominant species of macroalgae at each island. Additionally, to aid in the interpretation of coral microbiome responses, we also sampled microbes from turf, macroalgae and the water column.

We sequenced the 16S rRNA gene to characterize the prokaryotic microbial communities in our samples, and compared the microbial community composition, alpha diversity, beta diversity and the relative abundance of specific microbial taxa among groups to evaluate the local and regional effects of algae. Each of these metrics provided different insights into how the microbial communities changed for each sample type. We also compared the responses between the islands to evaluate if the resulting patterns were robust across reefs separated by approximately 1600 km and characterized by different communities, abiotic conditions, and levels of human impact. Contrary to previous hypotheses, we found that algae affected multiple aspects of coral microbiomes at both local and regional scales, and that the effects of algae at one scale modified their effects at the other scale. Here, we describe those patterns, potential mechanisms driving them, and their implications for our interpretation of previous studies exploring the effects of algae at only local scales.

## 2. Methods

### 2.1. Study locations

Field collections and surveys were conducted in reef sites around the islands of Mo'orea and Mangareva, French Polynesia (South Pacific) from 26 December 2017 to 21 January 2018. (All fieldwork and

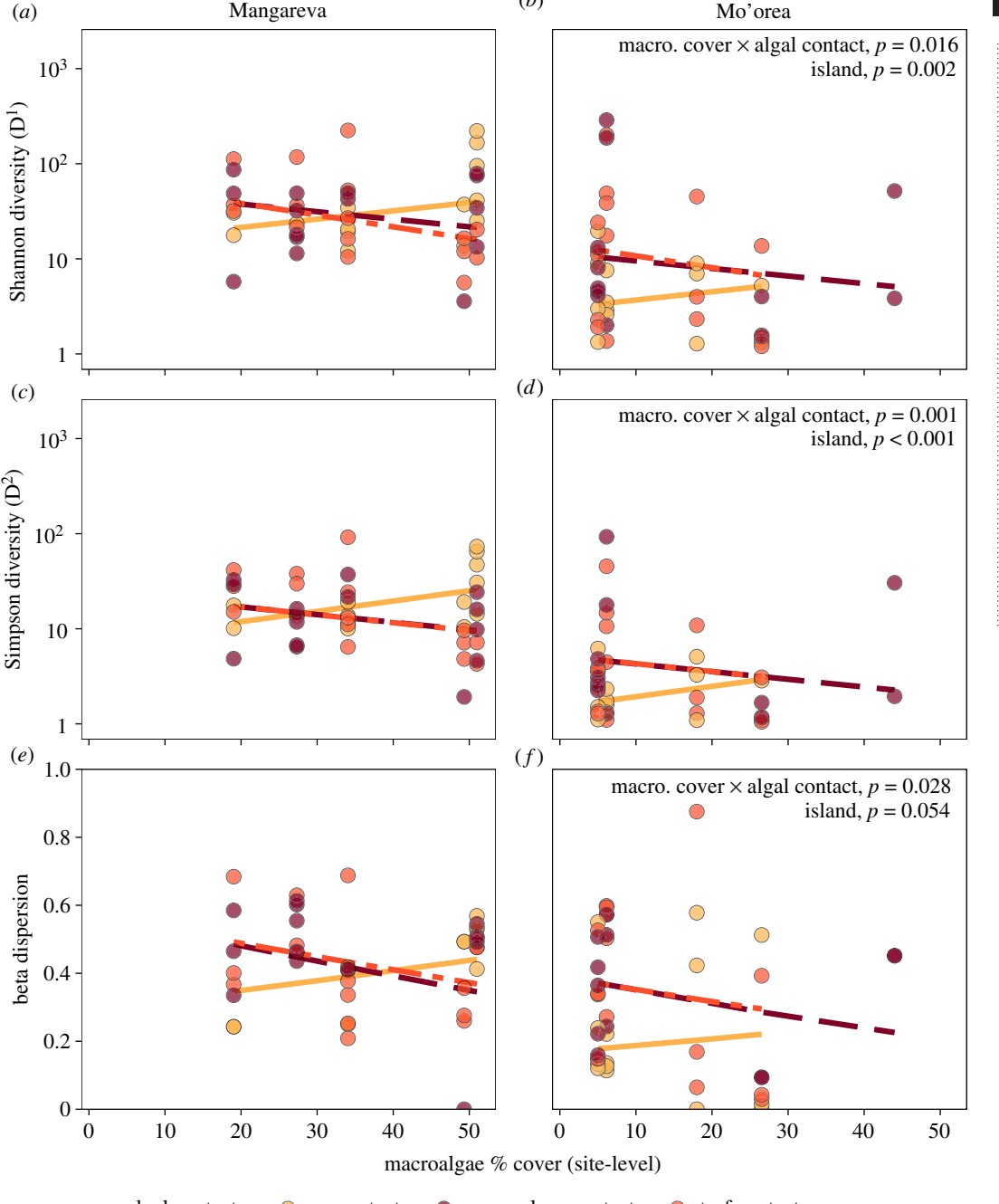

**Figure 2.** Coral microbial community (ASV-level) alpha and beta diversity versus site-level percent cover of macroalgae (macro. cover), for corals from each island (columns) and algal contact group (colours). (*a,b*) Hill number for *q* = 1, i.e. Shannon diversity. (*c,d*) Hill number for *q* = 2, i.e. Simpson diversity. (*e,f*) Beta dispersion. Points represent the observed values for individual coral samples, with predicted regression lines based on the estimated model coefficients backtransformed to the original scale. Statistically significant predictors are indicated for each response variable. (Significant main effects are not shown if the interaction is significant; results are summarized in electronic supplementary material, table S6). The *y*-axis for (*a–d*) is on a log$_{10}$ scale.

collections were approved by the French Polynesian government.) Mo'orea is located in the Society Islands archipelago, which is located approximately 1600 km to the northwest of Mangareva (Gambier Islands; electronic supplementary material, figure S2). Mo'orea has a larger human population (approx. 17 000) than Mangareva (approx. 1400) and regularly exchanges people and goods with the nearby (17 km), densely populated island of Tahiti [36]. Therefore, at least some of its reefs likely experience more anthropogenic influence than Mangareva.

## 2.2. Benthic surveys

At each island, we selected five back reef or fringing reef sites with an average depth of 1–2.25 m that encompassed a gradient of coral to algal dominance (electronic supplementary material, figure S2; GPS coordinates in electronic supplementary material, table S1). At each site, we haphazardly placed four 25 m transects (except in Mo'orea, where one site had five transects) parallel to shore, with each transect at least 10 m away from any other. We characterized the benthic community composition at each site by visually estimating the percent cover of all living and non-living substrates within ten 0.25 $m^2$ quadrats that were placed in pairs on each side of the transect (shore-side and ocean-side) at five fixed positions. We recorded the cover of any benthic organism (identified at the genus level) or non-living substrate (e.g. sand, rubble, newly exposed reef carbonate) that occupied at least 1% of the quadrat area. From these quadrats, we determined the mean site-level cover of turf, macroalgae, coral and other benthic functional groups. These site-level means were used as regional-scale predictors in our analyses.

## 2.3. Field microbial sampling

We identified the most widely distributed, abundant coral and macroalgal taxa at each island (i.e. the dominant taxa) for microbial sampling: the coral *Montipora aequituberculata* and the macroalga *Dictyota bartayresiana* at Mangareva, and the massive *Porites* species complex (including species *Porites lobata* and *Porites lutea* that are not visually distinguishable in the field [37,38]) and *Turbinaria ornata* at Mo'orea. At each site on both islands, we also sampled turf algae, i.e. short lawns (less than 1 cm) of primarily filamentous algae and cyanobacteria intermixed with small juvenile forms of macroalgae. Along each transect, we identified one coral individual of the dominant taxon in each of three local interaction types: (1) no contact, greater than 20 cm from any algae; (2) in contact with turf and (3) in contact with the dominant macroalgae for that island. Using a 4 mm biopsy punch, we collected a standardized amount of tissue from each coral. For corals with macroalgae contact, tissue was collected in the area where macroalgae made contact with coral; for corals with turf contact, tissue was collected 1–2 mm from the interaction zone with the turf; and for the no-contact corals, tissue was collected at least 20 cm from any algal contact. (Previous studies have found that changes in coral microbiomes due to algal contact generally disappear more than 5 cm from the zone of contact [32,33]). At sites with only four transects, we sampled a fifth coral in each local interaction type to increase replication ($n = 5$). For the coral specimens in contact with algae, we also collected a sample of the macroalgae or turf that was near the zone of contact. Macroalgae were gently placed in a sterile 50 ml Falcon tube underwater and sealed, while turf was sampled with a 50 ml sterile syringe that we used to suck water off the surface of the turf. We also collected 1–2 water column samples per transect (collected at least 1 m from the benthos) using a sterile 50 ml syringe ($n = 5$ per site). Immediately after collection, samples were put on ice until they could be refrigerated and processed. In the laboratory, the Falcon tubes containing macroalgae were shaken for 20 s and then the seawater in each tube was collected by a sterile syringe. Each syringe for turf, macroalgae and water samples was passed through separate sterile Sterivex filters (0.22 µm, Millipore). After processing, all samples were preserved in 1 ml RNA*later* and refrigerated until they could be taken to a laboratory and frozen for storage at −20°C until DNA extraction.

## 2.4. Molecular methods (DNA extraction, PCR and bioinformatics processing)

DNA was extracted from samples using Qiagen DNeasy PowerSoil kits per the package instructions (Qiagen, Germantown, MD) with bead-beating for 15 min. For the filter samples (macroalgae, turf and water), we opened the filter cartridge over a sterile Petri dish and cut the filter into fine segments using a sterilized scalpel. Then we added the filter to the bead tube and added twice the volume of Solution C1 (120 µl instead of 60 µl). Following DNA extraction, we used the guidelines from the Earth Microbiome Project 16S Illumina Amplicon Sequencing Protocol for library preparation [39], except we added 2 µl of DNA for each sample. Additionally, we used 50 µM mPNA clamps for the library preparation of algal samples to prevent the amplification of chloroplasts, which would overwhelm the downstream sequences [40]. We used the primer set with barcoded forward primer 515FB [41] and reverse primer 806RB [42]. The V4 region of the 16S rRNA gene was amplified in triplicate for each sample using the Phusion High Fidelity PCR Master Mix (New England Biolabs, Ipswich, MA). PCRs used the following protocol. First, samples were denatured at 94°C for 3 min.

Next, samples underwent denaturation for 45 s at 94°C (or for 10 s at 78°C for the algal turf and macroalgal samples, as part of the mPNA clamps procedure [40]), followed by primer annealing for 1 min at 50°C, and extension for 1 min 30 s at 72°C, repeated 35 times. The final elongation step was at 72°C for 10 min. Samples were then held at 4°C or refrigerated. Negative PCR controls were examined by gel electrophoresis on ethidium bromide-stained, 1% agarose gels to determine if there was contamination. Triplicate PCR products were combined and purified using MinElute PCR Purification Kit (Qiagen). Cleaned and concentrated PCR product (amplicon library) concentrations were measured on a Nanodrop 1000 or a Denovix DS-11 FX+ (Denovix,Wilmington, DE). A total of 240 ng of each amplicon library was pooled for sequencing on an Illumina MiSeq with paired 150-bp reads (v. 2 cycle format) at the University of Florida Interdisciplinary Center for Biotechnology Research.

Barcodes and primers were removed using cutadapt v. 1.8.1 [43]. We then completed bioinformatics in R v. 3.2.0. We used the DADA2 v. 1.16 [44] pipeline for quality filtering, error estimation, merging of reads, dereplication, removal of chimeras and selection of amplicon sequence variants, i.e. ASVs (filtering parameters are provided in electronic supplementary material, appendix S1). ASVs are similar to OTUs (operational taxonomic units); however, they provide exact sequences instead of sequences grouped together based on a similarity threshold [44]. We used the Silva reference database (v. 132) in DADA2 to assign taxonomy to the ASVs to the genus level [45].

The sequence table, taxa table, and metadata table were then imported into the *phyloseq* R package [46]. We removed all chloroplast and mitochondrial DNA sequences and filtered out samples with a read depth less than 100 before analysing the prokaryotic microbial communities in each sample. Community analyses and calculation of diversity metrics were performed on ASV- and family-level data in R using the *phyloseq* [46] and *vegan* [47] packages.

## 2.5. Diversity responses

We quantified alpha diversity using the first- and second-order Hill numbers ($D^1$, Shannon diversity and $D^2$, Simpson diversity). These metrics provide the effective number of taxa, i.e. the number of equally abundant taxa needed to produce a community of equivalent diversity as the observed system [48,49], although each Hill number weights the contributions of rare and abundant taxa to alpha diversity differently. Shannon diversity has similar sensitivity to rare and abundant taxa, while Simpson diversity is influenced more by abundant taxa [49,50]. We used these Hill numbers to evaluate alpha diversity because they are relatively robust to low sample sizes [51], variation in sampling depth, DNA amplification biases and rare taxa, which are all common in microbial data [52–54]. Additionally, when used in combination, Hill numbers provide information on the distribution of taxon abundances, with declines from $D^1$ to $D^2$ indicating heterogeneity, i.e. unevenness, in abundances [55]. We did not calculate the zero-order Hill number ($D^0$, richness), as we had a small proportion of samples with low sequencing depth (electronic supplementary material, figure S4).

Beta diversity, i.e. the amount of variability in the microbial communities among individual samples in a group, was quantified using the dispersion of samples in multivariate space. High beta dispersion is an indicator that the microbial communities in a group of samples have been destabilized [56,57], which could occur in response to a stressor such as algal contact. Using the relative abundance of microbial taxa (not rarefied), we calculated pairwise weighted Bray–Curtis dissimilarity among samples, and then used this dissimilarity matrix to estimate the distance from a sample to its group centroid. These distances were calculated using the *betadisper* function in the *vegan* package in R, with groups based on substrate type (coral, water, turf, macroalgae) and site (1–5 at each island), or site and local algal contact (no algal contact, turf contact or macroalgal contact) for the within-coral comparisons. For three instances, in which there was only one sample in a group (electronic supplementary material, table S2), we omitted that group from analyses.

We fitted linear mixed effects models to the alpha diversity data (Shannon and Simpson diversity) using the *lmer()* function in the R package *lme4* [58]. We used beta regression, which is appropriate for analyses of continuous proportions [59,60], to analyse the beta dispersion data. The coral samples contained several dispersions equal to zero, which cannot be fitted with beta regression, so we used the rescaling equation in electronic supplementary material, appendix S3 of [59] to add a small constant to all of the dispersions to remove the zero values. We fitted the beta regression models with a logit link function using the *glmmTMB* package in R [61]. To compare our alpha and beta diversity responses among substrate types (i.e. coral, turf, macroalgae, water) and islands, we fitted models that included island, substrate type, and their interaction as fixed effects, and site as a random effect. To identify the effect of macroalgal cover on the microbiomes of each substrate and if that effect differed by island, we

fitted separate models to each substrate type. Models for the algae and water microbiome samples included the fixed effects of island, site-level macroalgal cover (i.e. a regional effect of algae), and their interaction, and site as a random effect. Models for the coral samples included algal contact (no algal contact, contact with turf or contact with macroalgae) as an additional fixed effect. To test the hypothesis that algae have local and regional effects that interactively influence coral microbiomes, and to determine if that interaction was consistent between islands, we fitted models that contained a three-way interaction of algal contact × macroalgal cover × island to the coral diversity responses. When the three-way interaction was not significant, we then tested a model that contained algal contact × macroalgal cover and island. $p$-values for the fixed effects were estimated using a Wald $\chi^2$ Type III test, calculated using the *car* package in R [62]. Model residuals were checked for patterns that indicated violation of model assumptions.

We focused our analyses on the site-level effects of macroalgae, rather than other regional descriptions of algal communities (e.g. turf cover or total algal cover) for several reasons. First, previous studies have found that macroalgae release more labile forms of DOC and these exudates increase densities of bacterioplankton more than coral exudates [22,23]. Turf algae also release large amounts of labile DOC [10] but have substantially lower biomass per unit area than macroalgae. Additionally, turf cover was lower and less variable at our sites (electronic supplementary material, figure S3). Thus, when scaled up to the site level, this could result in smaller absolute effects of turf on seawater DOC and microbial growth than macroalgae, making it a less informative regional predictor. We evaluated the contribution of macroalgae to variation in the macroscopic benthic community composition among sites (relative to other benthic functional groups) using principal component analysis (PCA). We performed PCA on the centred and variance-standardized mean percent cover of the main benthic functional groups (coral, macroalgae, turf, crustose coralline and peyssonnelid algae, non-living substrate, and 'other') at each site.

## 2.6. Community composition

Due to the highly skewed nature of our data, which contained many rare ASVs and a few highly abundant taxa, we compared microbial community composition of our samples in ordination space using non-metric multidimensional scaling (NMDS), based on the weighted Bray–Curtis dissimilarity matrix, calculated from the relative abundance of microbial taxa within each sample at the ASV level. We also provide results at the family level, primarily in the appendix. We calculated the stress of each NMDS to evaluate how well it described the community composition. Each NMDS had stress less than 0.2 indicating adequate model fit to the data [63]. Differences in composition were tested using PERMANOVA (999 iterations), with the *adonis* function in *vegan* [47], fitting models containing substrate type, island, and their interaction. Because that analysis demonstrated that community composition differed between the islands (see §3.3), we fitted separate models to the Bray–Curtis dissimilarities of each substrate type at each island and used the *adonis2* function to perform a PERMANOVA to test the effects of the continuous predictor, site-level macroalgal cover. Coral models also included local algal contact and its interaction with site-level macroalgal cover as predictors. We visualized the changes in coral and macroalgal communities with site-level macroalgal cover using separate NMDS ordinations for each island. For each island, we binned site-level macroalgal cover into three categories: low (0–19%), medium (20–35) and high (greater than 35%), and created a separate ordination plot for the samples in each bin.

## 2.7. Identifying specific taxa that change

Finally, we used separate beta-binomial regression models for each island to identify which coral microbial families responded to site- or local-level effects of algae, implemented using the *corncob* package in R. This approach accounts for variation in sampling depth among samples, can handle zero counts, and is robust to other common issues in microbial data, such as overdispersion of abundances [64]. For each island, we modelled the effects of algae on the mean relative abundance of microbial families at each scale (local contact or site-level macroalgal cover), while controlling for the effects of algae at the other (non-focal) scale, and overdispersion differences between groups (i.e. differences in within-group variation). These analyses controlled for a false discovery rate of 0.05. Separate models were fitted for each island because of the large differences in the composition of the coral microbiomes between islands.

To evaluate whether the families that changed in response to algae were major constituents of the coral microbiome, and if the families that increased with algae potentially came from sources such as

algae or the water column, we characterized the most common microbial families associated with each substrate type. We determined which microbial families were present in at least 30% of samples and had a median relative abundance greater than 1% across all samples of a substrate type from each island (which we categorized as the dominant microbial taxa for that island–substrate combination). We then compared these dominant microbial families to the families that were identified by the corncob analysis of the coral microbial communities. If a family that increased in corals with algal contact or site-level macroalgal cover was a dominant family in the turf or macroalgae samples, this suggested that algae might transfer propagules of this taxa to the coral, and/or stimulated its growth within the coral community. If water-associated taxa increased in corals with algal contact or site-level macroalgae, this could suggest that algae disrupted the coral microbiome, allowing opportunistic invasion of microbial taxa from the surrounding environment.

# 3. Results

## 3.1. Site- and island-level patterns in benthic communities

Benthic community composition varied by site and island (electronic supplementary material, figures S2 and S3). Non-living substrates (e.g. sand, rubble) were present at all of the sites, but were more common in Mo'orea (site means ranged from 3 to 55% cover) than in Mangareva (2–30%), primarily because reefs in Mangareva were larger or more contiguous. Mean site-level macroalgal cover ranged from 5–44% at Mo'orea and 19–51% at Mangareva, while coral cover ranged from 5–36% at both islands (electronic supplementary material, figure S3). Turf algae ranged from 8–23% and 15–32% at Mangareva and Mo'orea, respectively, and cyanobacteria cover was generally low (0–2%). PCA indicated that sites were largely separated by variation in macroalgae and non-living substrate on the first principal component (PC1, which explained 45% of the variance; electronic supplementary material, figure S4). Macroalgae had the highest loading on PC1, indicating that it contributed the most to variation on this axis (electronic supplementary material, table S3). Coral and turf algae largely distinguished sites on the second component (PC2, which explained 24% of the variance; electronic supplementary material, figure S4, table S3). Thus, macroalgal cover seemed to be an appropriate variable to distinguish the benthic communities of our sites while also allowing us to directly test our hypotheses about the effects of algae on reef microbes.

## 3.2. Sequence characteristics

We had 8 239 519 total reads across all samples after quality filtering and removing samples with low read depth (fewer than 100 reads per sample). Samples with low read depth were distributed across all sites, islands, and sample types, although they were particularly prevalent among the coral samples and sites 2 and 5 at Mo'orea. After filtering and accounting for a handful of additional samples that were lost due to human or equipment error, we had 186 samples, generally with 2–5 samples per site and local algal contact or substrate type group (electronic supplementary material, table S2). Sequence data are available in the NCBI SRA database under BioProject PRJNA681520. Post-filtering reads per sample ranged from 101 to 204 285 reads, with a median of 31 859 and a mean of 44 298. Sampling depth for each substrate type is summarized in electronic supplementary material, figure S4 and table S4. Post-filtering ASVs per sample ranged from 7 to 2316, with a mean of 625 and a median of 590. ASV counts were heavily skewed, with most ASVs being rare (less than 100 reads across all the samples).

## 3.3. Variation among substrate types and islands

### 3.3.1. Diversity responses

Microbial community alpha diversity (measured as Shannon diversity, i.e. $D^1$, the Hill number for $q = 1$, and Simpson diversity, i.e. $D^2$, the Hill number for $q = 2$) and beta diversity (measured by beta dispersion) differed by substrate type (coral, macroalgae, turf, and water; for coral we used samples from all local contact categories) and island, with a significant interaction between island and substrate type (electronic supplementary material, figure S6a,b,c and table S5). Corals had approximately one half to one fourteenth of the alpha diversity observed on macroalgae, turf, and water (electronic supplementary material, figure S6a,b), although the magnitude of the difference between coral and

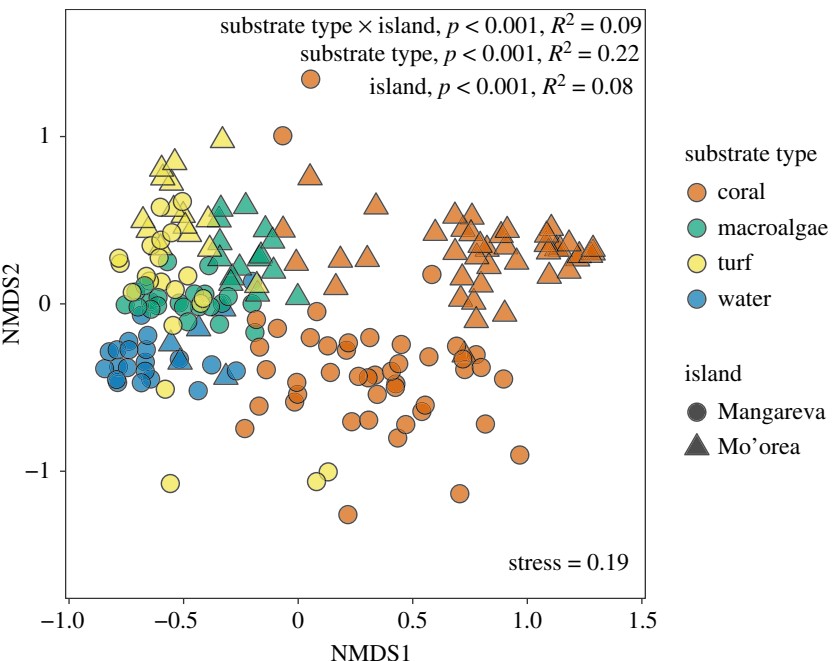

**Figure 3.** NMDS ordination based on weighted Bray–Curtis dissimilarity of microbial community composition (ASV-level) associated with different substrate types (colours) from two islands (shapes). Coral samples include corals from all three algal contact groups (in contact with macroalgae or turf, or not in contact with algae). PERMANOVA results are shown, with additional details in electronic supplementary material, table S8.

the other substrate types was generally larger at Mo'orea, and when alpha diversity was equally weighted by the contributions of both rare and dominant taxa (i.e. Shannon diversity). Beta dispersion was the lowest in the water samples (i.e. these samples were the most homogeneous), with similar dispersion in the water samples at both islands (electronic supplementary material, figure S6c). By contrast, the coral and algae samples had demonstrably higher dispersion, indicating a high degree of variability in the microbial communities of individual samples. These differences in dispersion between water and the other substrate types depended on island. At Mangareva, dispersion in the coral and turf samples was approximately twice the dispersion in water samples; macroalgae exhibited intermediate dispersion. At Mo'orea, coral, macroalgae and turf had similar dispersion, which was 1.6–1.8 times the dispersion among water samples.

### 3.3.2. Community composition

NMDS ordination indicated that the microbiome composition of samples tended to cluster by substrate type, with some separation between islands within a substrate type (figure 3). Generally, macroalgae, turf, and water samples were more similar to one another than to coral samples. Coral samples generally showed the greatest variability in microbiome composition (reflected in the spread of points in figure 3, with coral samples from Mangareva exhibiting the greatest variation). Some of this variation in the coral microbiomes might have been a result of aggregating coral samples across the three algal contact groups: e.g. some coral samples (especially from sites at Mangareva with high macroalgal cover) were close to algae in ordination space (see further analyses below). PERMANOVA results indicated there was a significant interaction between the effects of substrate type and island on ASV- and family-level microbial community composition ($p = 0.001, 0.001, R^2 = 0.09, 0.10$, respectively; electronic supplementary material, table S8), largely due to the greater differentiation in the coral microbiomes between the two islands compared to the other substrate types.

### 3.3.3. Dominant families

Although there was a large amount of overlap in the families found across all substrate types, the identity and relative abundance of the dominant microbial families generally differed among substrates, and to a lesser extent, islands within a substrate (electronic supplementary material, appendix 2, figures S9 and S10).

More dominant microbial families were shared between macroalgae, turf, and water than were shared with corals. Within a substrate type, dominant families differed the most between islands in the coral samples. *Montipora* corals from Mangareva contained a total of 416 microbial families, but had 10 dominants: Alteromonadaceae, Colwelliaceae, Endozoicomonadaceae, Flavobacteriaceae, Midichloriaceae, Pseudoaltermonadaceae, Rhodobacteraceae, Saccharosprillaceae, Saprospiraceae and Vibrionaceae. Community dominance was distributed relatively evenly among these families, with Pseudoaltermonadaceae as the most common family (electronic supplementary material, figure S9). By contrast, massive *Porites* corals from Mo'orea were overwhelmingly dominated by Endozoicomonadaceae, which comprised on average 65% of the microbial reads observed in these samples (with a median of 77%). Vibrionaceae was the only other family meeting the dominance criteria in Mo'orea corals (out of 420 families). Dominant families in the algae and water samples are described in electronic supplementary material, appendix 2 and figure S10.

## 3.4. Variation associated with site-level macroalgal cover and algal contact

### 3.4.1. Diversity responses

Within the coral samples, there was a significant interaction between local algal contact and site-level macroalgal cover for both Shannon diversity and Simpson diversity ($p = 0.016$, $p = 0.001$; electronic supplementary material, table S6). Shannon ($D^1$) and Simpson diversity ($D^2$) of coral microbiomes increased with site-level macroalgal cover in corals that were *not* in local contact with algae (figure 2a–d). In contrast, Simpson and Shannon diversity slightly declined with site-level macroalgal cover in corals that were in contact with macroalgae or turf (figure 2a,e). Thus, algal contact caused alpha diversity to increase in corals when site-level macroalgae cover was low (below approx. 30%) and decrease when site-level cover was high (above approx. 40% cover), when both considering rare and dominant taxa (Shannon diversity), versus primarily dominant taxa (Simpson diversity). In addition, both alpha diversity metrics differed between islands ($p = 0.002$, less than 0.001, respectively), with corals from Mangareva having higher microbial alpha diversity than corals from Mo'orea. In contrast to coral microbiomes, the alpha diversity of microbial communities associated with macroalgae, turf and water was unresponsive to variation in site-level macroalgal cover (electronic supplementary material, figure S7 and table S7).

Beta dispersion, or variability in the microbial communities among individual samples, generally responded similarly to alpha diversity (electronic supplementary material, tables S6 and S7). For the coral samples, there was a significant interaction between local algal contact and site-level macroalgal cover ($p = 0.028$). Beta dispersion trended lower in corals from Mo'orea ($p = 0.054$), but increased with site-level macroalgae cover for corals that were not in contact with algae at both islands (figure 2e,f). Dispersion declined with site-level macroalgae in corals contacting turf and macroalgae. In contrast to the coral samples, microbial beta dispersion in macroalgae, turf, and water column samples did not respond to site-level macroalgal cover (electronic supplementary material, figure S7 and table S7).

### 3.4.2. Community composition

Coral microbiome composition responded differently to site-level macroalgal cover and local algal contact at each island (electronic supplementary material, table S9). At Mangareva, coral microbiomes shifted with macroalgal cover at (PERMANOVA: $p = 0.002$, $R^2 = 0.07$) when compared at the ASV level. However, family-level analyses of microbiomes from Mangareva showed interactive effects of algal contact and macroalgal cover ($p = 0.03$, $R^2 = 0.07$). By contrast, macroalgal cover and algal contact were not associated with significant changes in microbiome composition in corals from Mo'orea (electronic supplementary material, table S9).

The microbiomes of macroalgae (ASV-level) were associated with site-level macroalgal cover at both islands. However, family-level analyses of microbiomes of macroalgae only varied with macroalgal cover at Mo'orea (electronic supplementary material, table S9). Turf microbiomes (ASV- and family-level) varied with macroalgal cover at Mo'orea, but not at Mangareva (electronic supplementary material, table S9). By contrast, water microbiomes did not vary with macroalgal cover at either island when compared at the ASV- or family-level (electronic supplementary material, table S9).

Viewed in the same NMDS ordination space, the microbiome composition of coral samples tended to become more similar to the macroalgae samples as site-level macroalgae increased at Mangareva (figure 4a–c). This was especially obvious for corals not in contact with algae (orange circles) along

**Figure 4.** NMDS ordination based on Bray–Curtis dissimilarity of ASV-level microbial communities associated with macroalgae (green triangles) and coral (orange, red circles) samples from Mangareva (*a–c*) and Mo'orea (*e–g*), compared across different levels of site-level macroalgal cover: low (0–20%), medium (20–35%) and high (greater than 35%). Local algal contact groups for the coral samples are indicated by the colour of each circle. Macroalgal bins are used here for visualization purposes only; PERMANOVA used continuous macroalgal cover values (*p*-values on graph, full results in electronic supplementary material, tables S9 and S10). (*d,h*) The position of samples along NMDS1 (the major axis distinguishing coral from macroalgae samples) versus site-level macroalgal cover (*p* values for significant predictors on graph, full results in electronic supplementary material, table S11).

the primary axis distinguishing the coral from the macroalgae samples (NMDS1; figure 4*d*). However, this effect was much weaker at Mo'orea, which generally had lower macroalgal cover and fewer samples (figure 4*e–g*). To test these changes in the similarity of coral versus macroalgae microbiomes, we fitted linear mixed effects models with site as a random factor and macroalgal cover and sample-contact type (either macroalgae, coral-no contact, coral-turf contact, or coral-macroalgae contact) as fixed-effect predictors of NMDS1 in the coral and macroalgae samples at each island. Mangareva samples demonstrated a significant macroalgal cover×sample-contact type interaction (*p* = 0.024; electronic supplementary material, table S11), with the position of coral samples on NMDS1 becoming more negative (and, thus, closer to the macroalgae samples) as macroalgal cover increased, for corals with no algal contact and macroalgal contact (figure 4*d*). Corals with turf contact did not move closer to macroalgae samples on NMDS1 as macroalgal cover increased. In contrast, at Mo'orea, macroalgal cover was not an important predictor of the position of coral and macroalgae samples on NMDS1, although the sample-contact types did significantly differ (*p* < 0.001; electronic supplementary material, table S11). Corals with no algal contact were the farthest from the macroalgae samples on NMDS1, while corals with turf or macroalgae contact were closer on average (figure 4*h*). (However, these results must be interpreted cautiously, as NMDS ordination distances are not independent.)

### 3.4.3. Responses of specific microbial families

We used corncob analysis to identify which microbial families changed in coral samples in response to algal contact and site-level macroalgal cover at each island, and to estimate the magnitude of those effects (figures 5 and 6). We also compared the identified families to the dominant microbial families found on corals and other substrates (electronic supplementary material, figures S9 and S10) to explore potential mechanisms through which coral microbiomes changed in response to site-level algae, such as through the transfer of common algal microbes and their subsequent increase in the coral microbiome.

Within the coral microbial communities, the majority (85% or 34/40) of microbial families that were identified by the corncob analysis as changing in response to algae did so at only one scale (either in

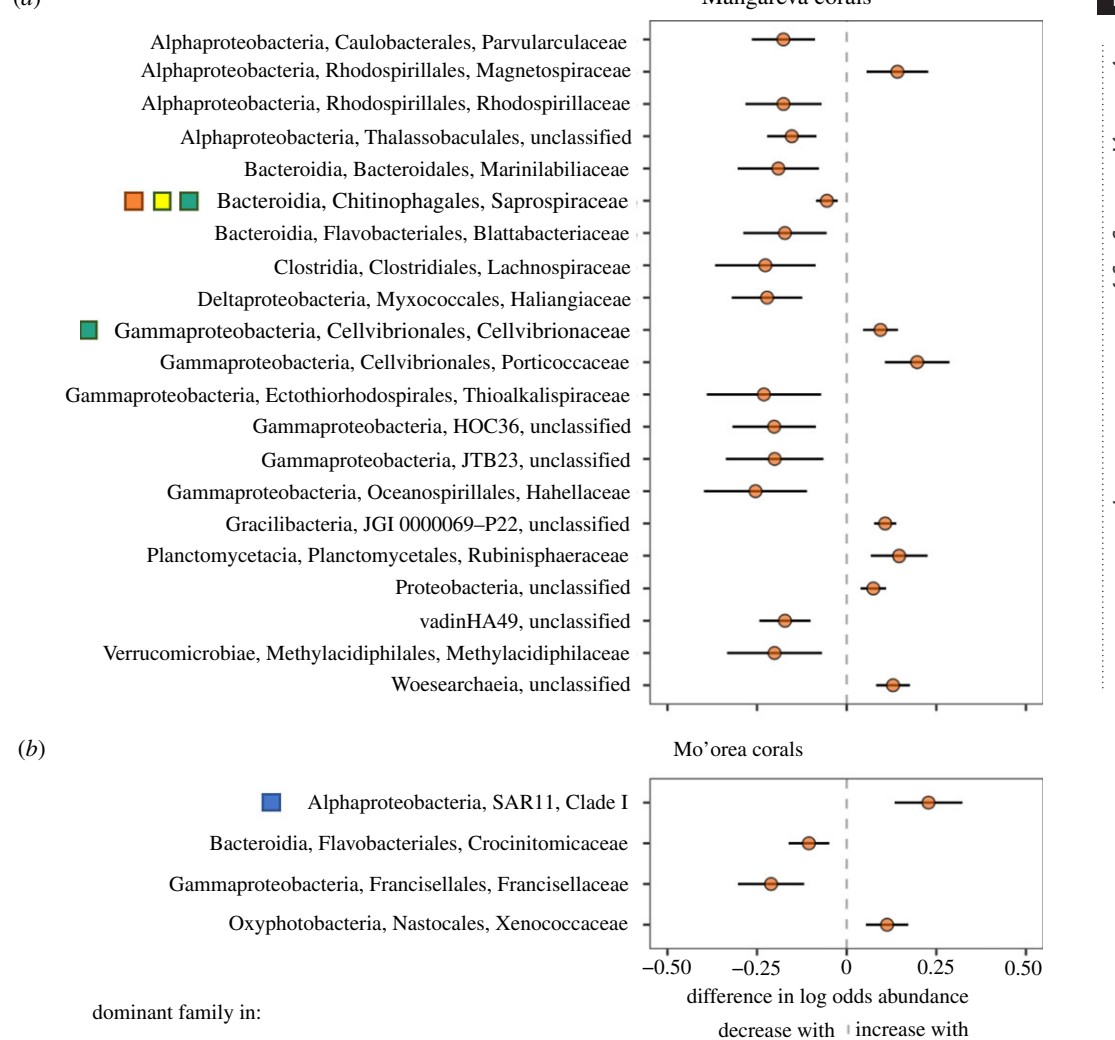

**Figure 5.** Microbial families in the coral samples that had changes in abundance associated with site-level macroalgal cover, controlling for the effect of local algal contact, at (*a*) Mangareva and (*b*) Mo'orea. Positive numbers indicate taxa that increased with site-level macroalgal cover, negative numbers indicate taxa that declined with macroalgal cover. Points represent coefficient estimates for each taxon; error bars represent 95% CI. Squares identify microbial taxa that were a dominant family in one or more of the substrate types (coral, turf, macroalgae, or water, indicated by the square colour). Taxa are organized alphabetically by class, order, family.

response to algal contact, or in response to site-level macroalgal cover). Only 6/40 families changed in response to both factors. Additionally, the families that responded to algae at each scale differed between islands. Within an island, most families (67 and 75% at Mangareva and Mo'orea, respectively) exhibited the same directional response to contact with turf versus macroalgae, although approximately one third (5 out of 15) of the responses for one algal contact group were not significant (as inferred from their 95% CI intervals). Four of the 6 families that responded to algae at both scales responded in the same direction at both scales.

Most of the families identified in the corncob analysis as showing responses to algae were not dominant members of the corals' microbiomes. The notable exceptions were Endozoicomonadaceae and Saprospiraceae, which declined with algal contact and site-level macroalgal cover, respectively (figures 5*a* and 6*a*). However, 33% of the families that increased with either algal contact group (turf or macroalgae) or with macroalgal cover were a dominant family in either the turf or macroalgae microbiomes for that island, e.g. Cellvibrionaceae, Cyclobacteriaceae, Rubritalaceae, Nostocaceae, and Hyphomonadaceae (figures 5 and 6), suggesting that algae were a source of these microbes. Additionally, several oligotrophic taxa that were dominants in the water samples declined with algal

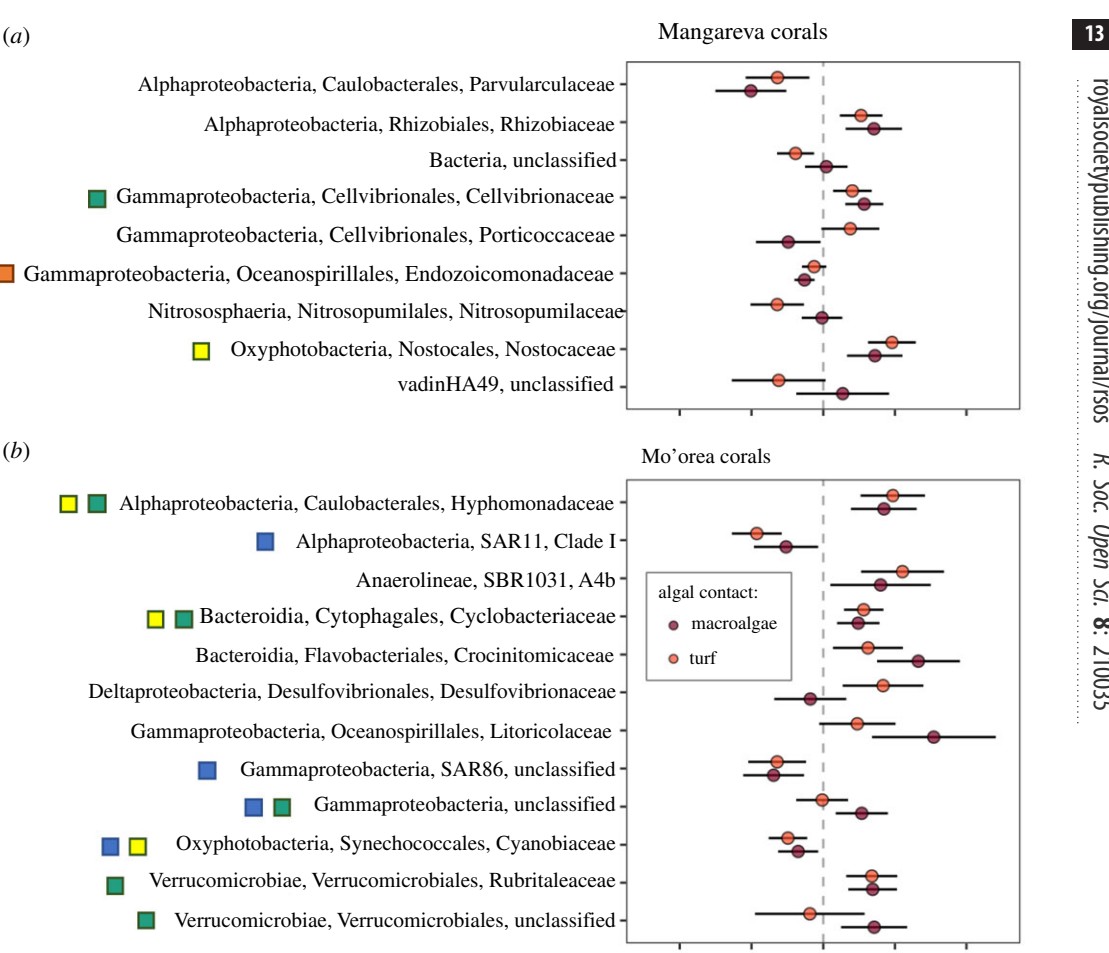

**Figure 6.** Microbial families at each island that changed in response to local algal contact, controlling for the effect of site-level macroalgal cover. Changes in taxa are shown for coral in contact with macroalgae or turf (indicated by the colour of the points) relative to coral samples not in contact with algae. Positive numbers indicate a family increased with algal contact, negative numbers indicate a family declined with algal contact. Points represent coefficient estimates for each taxon; error bars represent 95% CI. Squares identify microbial taxa that were a dominant family in one or more of the substrate types at that island (coral, turf, macroalgae, or water, indicated by the square colour). Taxa are organized alphabetically by class, order, family.

contact at Mo'orea (e.g. SAR11 SAR86, and Cyanobiaceae, the latter of which contained sequences from the genera *Synechococcus* and *Prochlorococcus*).

# 4. Discussion

Previous studies have observed within-species variation in the microbiomes of corals from different geographic locations [65,66]. Our results suggest that site-level macroalgal cover could help explain those differences. Coral microbiomes showed shifts in alpha and beta diversity with site-level macroalgal cover, with the direction and magnitude of these changes depending on local algal contact. Specifically, when corals were in contact with algae, the alpha and beta diversity of their microbial communities declined with site-level macroalgal cover (for alpha versus beta diversity, respectively). In contrast, alpha and beta diversity increased with site-level macroalgal cover in corals that were *not* in contact with algae. This statistical interaction was more extreme than a simple antagonism: it caused the direction of the effects of algal contact to switch from negative to positive at different levels of site-level macroalgal cover (figure 2). These results indicate that corals free from local competitive interactions with algae could still be affected by algae present elsewhere at a reef site. They also reveal that it is difficult to predict how coral microbiomes will respond to competitive

interactions with algae without also considering algal densities at a larger (e.g. site-level) spatial scale, which could contribute to why studies have reported increases [19,56], decreases [18,67], and no effect [68] of algal contact on coral microbiome alpha and beta diversity.

The interactive effects of site-level macroalgal cover and local algal contact on coral microbiome alpha diversity (figure 2a–d) could be the net effect of several processes that potentially increase the establishment of new microbes in the coral microbiomes and/or alter the growth of specific microbes within the microbiome. For example, if the densities of microbes in the water column increase as site-level algae increase (as in Haas *et al*. [22]), this could increase the rate at which microbes colonize corals. If this immigration rate was sufficiently high, it could overwhelm the ability of corals to regulate their microbiome, leading to the addition of new taxa and thereby increasing the alpha diversity of the coral microbiome. Conversely, algae could change abiotic conditions (e.g. increase dissolved organic material or allelochemicals, or reduce pH, $O_2$, etc.), and this could increase the growth of some microbial taxa (particularly copiotrophs and potential pathogens), and decrease the growth of others [22,23,69]. When these changes in abiotic conditions are large enough, such as when corals are in contact with algae at sites with high algal cover, fast-growing, algae-tolerant microbes might be able to proliferate and displace taxa associated with the healthy coral microbiome, thereby reducing microbiome alpha diversity. Therefore, the antagonistic effects of local and regional algae on coral microbiome alpha diversity are likely a balance between diversity-decreasing processes associated with altered microbial growth and diversity-*increasing* processes associated with enhanced colonization. Furthermore, because increased site-level macroalgal cover generally did not increase alpha diversity for corals contacting algae (figure 2a–d), our results suggest that algal contact alone saturates the diversity-increasing process associated with colonization.

Changes in coral microbiome beta diversity mirrored those observed for alpha diversity (figure 2a–d versus 2e,f). This correspondence between alpha and beta diversity suggests that the taxa that caused increases in alpha diversity were variable among individual corals from the same site and algal contact group. Moreover, as site-level macroalgal cover increased, some coral microbiomes shifted towards the microbial community composition of the algae samples, while other coral microbiomes did not respond or shifted towards an entirely different community configuration, accentuating the among-coral variability (figure 4). Local algal contact also tended to make coral microbial communities more variable (figure 2e,f), with some coral microbiomes becoming more like the algal microbial communities (figure 4). However, algal contact and site-level macroalgal cover exhibited antagonistic effects, much like they did with alpha diversity (figure 2e,f). This antagonistic effect of local algal contact and high site-level macroalgal cover on beta diversity could occur through a similar mechanism as we proposed for alpha diversity. Specifically, increases in microbial densities due to algae might facilitate opportunistic colonization of the coral microbiome by new taxa, with successful colonists differing among individual corals (thereby increasing beta diversity). However, large shifts in abiotic conditions created by algal contact *and* high site-level algal cover could reduce the number of microbial taxa capable of persisting in the coral microbiome, causing the microbiomes of individual corals to become more similar to one another. Independent of the mechanism, our results support the conclusion that algae tend to shift coral microbiomes to be more similar to algal microbiomes [17], although a coral's microbiome response to algae can be highly variable [19,56].

Together, our results suggest a new model of algal effects not anticipated in figure 1. Assume that we can convert the exposure of corals to algae by combining the local and regional effects into a single aggregate measure of exposure. Our results then suggest a unimodal (hump-shaped) relationship between coral microbiome diversity and the coral's exposure to algae (figure 7). The initial increasing part of this curve could represent the range over which enhanced microbial colonization due to increasing algal exposure dominates the external processes driving microbiome diversity. Corals with no algal contact but with low to high regional algal cover could fall within this range. In contrast, the declining part of the curve could represent the algal exposure over which fast-growing taxa proliferate and begin to take over the coral microbiome, such as for corals with algal contact and/or from sites with very high macroalgal cover. This relationship could generate the antagonistic patterns of interaction we observed between local and regional effects. The slopes of the increasing and decreasing portions of this curve and the location of the peak could depend on a coral taxon's innate ability to modulate its microbiome and/or resist invasion of new microbial taxa, as well as site-specific conditions such as water flow, nutrient levels, and other factors that might influence microbial dispersal and colonization as well as species sorting processes within the microbiome (e.g. growth, competition, and survival).

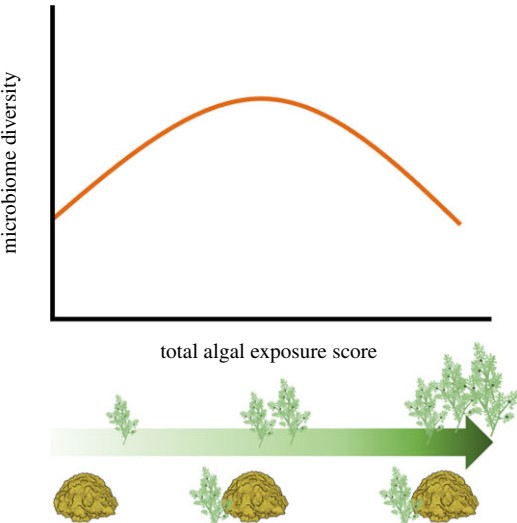

**Figure 7.** Hypothetical response of microbiome diversity (alpha or beta) to the overall exposure of corals to algae at both local (e.g. contact) and regional (e.g. site-level algal cover) spatial scales (total algal exposure score). Corals at sites with low macroalgal cover and not in contact with algae have the lowest exposure, while corals in contact with algae at sites with high algal cover have the highest exposure score.

## 4.1. Specific microbiome changes

Although there was variability in the microbial community composition of corals, some taxa changed predictably with site-level macroalgal cover or algal contact. Generally, site-level macroalgal cover caused smaller changes in the relative abundances of specific microbes compared to the effects of contact with algae at both islands (figure 5 versus figure 6). Additionally, fewer of the microbial taxa that increased with site-level macroalgal cover were common in the algae samples (approx. 11%), whereas approximately half of the families that increased in a least one algal contact group were dominant families in the turf or macroalgae samples. One explanation for this pattern is that direct algal contact might transfer more microbes to corals than are transferred via the water column, or more greatly change abiotic conditions and/or increase the concentration of the algal by-products that influence microbial growth. This could occur because algae-associated microbes and by-products are likely diluted in the water column relative to the interaction zones between corals and algae. Additionally, biotic and abiotic filtering in the water column could further reduce densities of algae-associated taxa before they colonize corals. Together, these processes could reduce the site-level effects of algae relative to the direct local effects of algal contact.

Several families that declined in response to either local or site-level algae were potentially beneficial symbionts that have been suggested to have coevolved with corals, including Hahellaceae and Endozoicomonadaceae (order Oceanospirillales), and Haliangiaceae (order Myxococcales) [15]. Hahellaceae and Endozoicomonadaceae are believed to be important for nutrient acquisition and cycling within corals [70], and have previously been found to decline in stressed corals [34,71]. Haliangiaceae is in an order of predatory bacteria that is correlated with disease resistance in corals [72]. Interestingly however, Endozoicomonadaceae declined in *Montipora* from Mangareva but not in *Porites* from Mo'orea, in which it was the most abundant microbial family (electronic supplementary material, figure S9). This could indicate a tighter relationship between the coral host and this microbial taxa in *Porites* compared to *Montipora*, making it more resistant to change. Other families of potential importance to the coral holobiont that declined with regional algal cover included Thalasobacculales and Rhodospirillaceae, which are positively associated with certain species of Symbiodinaceae [73]. Rhodospirillaceae is considered oligotrophic, and a previous study found that water column Rhodospirillaceae was associated with site-level coral cover [22]. Therefore, Rhodospirillaceae might decline in the water column as increasing regional algal cover increases water column DOC [22], which could subsequently reduce its incorporation into the microbiome of corals. Several other oligotrophic taxa that were dominants in the water samples (SAR11 Clade I, SAR86 and Cyanobiaceae) also declined with algal contact (figure 6), suggesting a similar mechanism related to DOC produced by algae.

As these potentially beneficial taxa declined in the coral microbiomes in response to local algal contact or increasing regional macroalgal cover, other taxa increased. Increasing taxa included five of the most abundant microbial families found in the algae samples (Cellvibrionaceae, Cyclobacteriaceae, Rubritalaceae, Nostocaceae and Hyphomonadaceae), suggesting algae acted as a source for these taxa. Other families that increased with local or site-level algae included taxa that are enriched in corals stressed by disease or environmental factors, including several associated with stony coral tissue loss syndrome (Rubritalaceae, Cyclobacteriaceae and Rhizobiaceae) [74,75], and one associated with black band disease and elevated water temperature (Desulfovibrionaceae) [76–78]. Desulfovibrionaceae and Rhizobiaceae are common in marine sediments [75]. Since both turf and macroalgae readily capture sediments [79–81], algae may encourage the introduction of sediment-associated microbes to coral microbiomes.

Our results indicate that functionally and taxonomically distinct types of algae can have qualitatively similar effects on coral microbial communities. Both macroalgae and turf had local effects on coral microbiomes, shifting their alpha and beta diversity and the abundance of many microbial families in similar directions. Site-level macroalgal cover also influenced diversity and composition of microbiomes. However, results were not consistent between islands for all microbiome responses. For example, site-level macroalgal cover had weaker effects on microbiome composition at Mo'orea relative to Mangareva (figure 4). This difference could have been caused by the smaller range of site-level macroalgal cover at Mo'orea (and lower statistical power due to the loss of many samples from the highest macroalgal cover site). Additionally, differences in the coral and algae species at both islands could have influenced the magnitude of the response. Some research suggests that microbial communities in *Montipora* vary more across environmental gradients than microbial communities in *Porites* [82]. Similarly, previous work has found that *Dictyota* has stronger local effects on coral microbiomes than other fleshy algae [19]. Thus, we might have expected that the *Montipora* corals sampled at Mangareva would show larger effects of local and regional algae than *Porites* corals at Mo'orea, especially for corals in contact with macroalgae. However, since both macroalgae and coral species differed at each island, we cannot disentangle how the identities of each of these competitors affected the magnitude of the observed responses. Despite the lack of consistency in the response of microbiome composition between islands, alpha and beta diversity showed similar patterns between islands (figure 2), suggesting that the interactive effect of local and site-level algae on these diversity metrics (e.g. figure 7) might be robust across many coral and algae combinations.

# 5. Conclusion

Overall, our results indicate that algae alter coral microbial communities at both local and site-level scales, altering their community composition, alpha and beta diversity, and disrupting important microbial associations with putatively beneficial symbionts, while simultaneously promoting disease and stress-associated taxa. Despite these changes, none of the corals that we sampled were visibly diseased or unhealthy. The greater-than-antagonistic effects of local and regional algae on coral microbiome alpha and beta diversity agree with other multi-stressor studies on corals [83,84], which suggests that coral microbiomes might be constrained in the amount to which they can change without resulting in serious negative health consequences for the host. Thus, although shifts in microbiome community characteristics like alpha and beta diversity may be indicative of stress, it may be difficult to use these responses as direct estimates of the total amount of stress that a coral is experiencing, particularly when multiple stressors are simultaneously affecting the coral.

Many of the microbes that we observed changing are known to have ecological and physiological importance to the coral. As a result, our work suggests that algae could alter coral microbiome function, and therefore modify the resilience of corals to environmental stressors. Both local algal contact and increased regional algal cover appeared to make coral microbial communities more similar to the communities on algae, and algal contact increased the abundance of many dominant algae-associated microbes in corals, while simultaneously reducing the abundance of several beneficial microbial symbionts. These changes could reduce the beneficial functions performed by the coral microbiome, making corals more vulnerable to competition with algae and potentially less resilient to additional environmental stressors. Theoretical and empirical work suggests that coral reefs might exhibit bistability of coral versus algal-dominated states partially mediated by the strength of coral–algal competition [85–87]. Consequently, the microbiome shifts that we observed could result in stronger feedbacks between corals and algae than expected, reducing the size of the perturbation

needed to shift a reef to an algal dominated state and increasing the likelihood of bistability in coral–algal systems, making it more difficult to return to a coral-dominated state once a reef switches to algal dominance. It is important, however, to keep in mind that our work was observational; therefore, the patterns that we documented could have been caused not by algae, but by factors that were correlated with algal cover or algal contact. Future experimental work is needed to examine causation, unravel the interactions between local and regional algae on microbial community functions, and determine whether there are specific thresholds of regional algal cover or coral microbial community change that result in predictable shifts in coral health.

Ethics. A.A.B. and A.L.B. acquired foreign-researcher protocols to conduct the fieldwork and sample collections for this project that were approved by the Delegation of Research and the High Commission of the Republic of French Polynesia.

Data accessibility. DNA sequences: Genbank Bioproject PRJNA681520. The data and code used for all graphs and statistical analyses for this study can be found at the Dryad Digital Repository: https://doi.org/10.5061/dryad. k0p2ngf8b [88].

Additional figures, tables and appendices are provided in electronic supplementary material [89].

Authors' contributions. A.L.B. conceived this study. A.L.B. and A.A.B. designed it with contributions from C.W.O. A.L.B. and A.A.B. conducted the fieldwork. A.L.B. conducted the laboratory work and bioinformatics. A.A.B. analysed the data, interpreted the results and wrote the manuscript with critical revisions from A.L.B. and C.W.O.

Competing interests. We declare we have no competing interests.

Funding. This research was supported by the University of Georgia, Odum School of Ecology's Small Grants programme and the University of Florida's John J. and Katherine C. Ewel Fellowship.

Acknowledgements. We thank the government of French Polynesia for research permissions. Additionally, we thank Julie Meyer at the University of Florida for laboratory space and sequencing support, Jacques Soo and staff at the UC Berkeley Richard B. Gump Research Station for field support in Mo'orea, and Patrick Schmack and Pension Maroi for support in the field in Mangareva. Finally, we thank the reviewers and editor for their helpful and constructive comments.

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
