## [Peer Review File · Royal Society Open Science]

Review History

RSOS-210035.R0 (Original submission)

Review form: Reviewer 1

Is the manuscript scientifically sound in its present form?

Yes

Are the interpretations and conclusions justified by the results?

Yes

Is the language acceptable?

Yes

Do you have any ethical concerns with this paper?

No

Have you any concerns about statistical analyses in this paper?

No

Recommendation?

Accepts with minor revision (please list in comments)

Comments to the Author(s)

Briggs et al. use 16S microbial data to test how coral microbiomes shift in response to changes in algal cover (across regional scales) and intimate contact between corals and algae or its absence. The authors present several well articulated hypotheses to explain their results and some nice conceptual figures as well. Ultimately the authors find that many metrics used to study coral microbiomes (alpha diversity, beta diversity) increase with macroalgae community %cover whereas the opposite is observed when corals are in close contact with algae AND algae % cover is increasing. These results suggest that changes in coral microbiomes are both a function of intimate contact with algae, community macroalgae cover, and their interaction. This work, therefore, contributes to our understanding of coral microbiomes and the species-interactions (as well as community traits at varying scales) that lead to shifting microbial community composition.

Overall I found the paper well written, the experiment executed correctly and with appropriate statistical analyses (hats off to you! It is quite impressive). I would rate my comments as minor and seeking clarity overall.

Specifically, I find the terms "local effect and regional effect" to be confusing, as local to me implies a small spatial scale but not immediate proximity (i.e., touching). I don't want to discourage the authors from using the terms they want, but to me describing these terms, showing term fidelity, and re-emphasizing the "intimate contact" or proximity is key here. Similarly, the authors use both "site" and "regional" to describe larger spatial patterns across islands. I think again some term fidelity and using appropriate terms to best describe these effects when discussing them would be helpful. This need for clarity in terms and describing effects is, in my opinion, the most significant issue the manuscript faces, although this is easily addressed.

The results section is quite dense, but perhaps not overly so considering the amount of data to unpack. Perhaps some streamlining could be undertaken, but I appreciate if this cannot be done. I also appreciate the author's use of the supplement to provide further information on their study and its findings, which helped provide a lot of valuable information. In terms of data accessibility, a suggestion: it might be prudent to look into a 3rd party site to host the code/ data (such as zenodo or Dryad, which I believe has a deal with RSOS). I see these files in the supplement, but a standalone repo would allow for the code (R project/Rmd/data) to be stored in a succinct directory making it easier for others to access and reproduce analyses.

Well done and I look forward to seeing this paper published.

Specific comments.

Line 12: spatial scales – for clarity, I think the authors mean “are these effects context dependent based on algal density”, which may not necessarily be spatially explicit but rather determined by a across a range of species-interaction. Perhaps rephrase to emphasize the 2 levels you are focusing on (1) are micorbial interactions varying across spatial scales, (2) and does this relate to macroalgae/turf density. I think you say this quite elegantly in Line 88.

Line 14: colon in stead of comma? (same at Line 90). Could flip to say Moorea and Mangareva, FP (country last – as you do in Line 115). Also, here and elsewhere you have an apostrophe in Moorea (Mo'orea), when it should be an okina (Mo'orea).

Line 20-21: what is “local algae”? density? In Line 21 perhaps rephrase: coral microbiomes are affected by both site-level algae density (in absence of direct contact) and proximity effects between coral and algae. As written the take home isn't punching through.

Line 26-28: “Coral reefs” (many different types of ‘reefs’). Citation for the slip-to-slime, as it were?

Line 36/39: hereafter “microbiome?” To orient the reader to the term...

Line 54: is this to say the size of a site or the distance between sites? I also see the reference to “a local effect” in line 53 with the “local scales” in line 58 to be confusing. There is the local neighborhood (<1000 m²?) and the local neighbor (touching you in this case). Maybe there is a way to disentangle this semantically (i.e. how you discuss the “zone of interaction” in Line 61).

Line 59: citation to clarify this passage?

Line 63: can you again remind us what the distinction is between local and regional SITE scales? You describe this well in Line 344 heading as “local contact vs. site-level effects”. I suppose you are also discussing within site and across a region here too (or is site = region? Seems Line 370 is stating this is the case...)? Some clarification would help.

Line 68-75: rephrase: (line 68) algae effects on coral microbiomes increase with regional density of algae? In general I find it hard to follow when “algae effect” is stated but what it is affecting is not – i.e., “algae effects on coral microbiomes...” (as in Line 73). Specifying the in-contact (<5cm?), local (<1000m²), region (>1000m²?) is important earlier to make this statement in Line 75 clear.

Line 108: last part of sentence here is not clear.

Line 119: I’d be cautious here, as these effects on islands are very often modified around the island. i.e., more human impact in Cook’s Bay with high sewage or groundwater discharge? This would be low-to-absent on south shore...

Line 138: missing a “)” after refs.

Line 159: use the micro symbol

Line 190: no italics on reference (and again at line 211, 246)

Line 198: citation as [44-46]? I’d make sure refs all formatted correctly at final version, such as here and Line 213 “[50,51]”

Line 223: “with” perhaps instead of “and” site as a random factor. New paragraph at the start of Line 223 sentence?

Line 253: “no-local scale”? or what does non-focal mean here?

Line 255: “0.05”

Line 309 and elsewhere: there should always be a zero before a decimal place. Please correct throughout.

Line 355: the “like it did” and “narrowly missed the cutoff” seems a bit too informal. Clarify the effect you are referencing and rephrase non-significant results as a trend?

Line 383: be consistent with or without spacing before “%”

Line 410: comma after e.g.

Line 412: algae density at regional scales? Coral microbiomes responding to this or something else (a different reef substrate)?

Line 441-442: I don’t see the phrasing here as making the point more clear. Can this be clarified? Perhaps even simply putting “switch from negative (in contact with algae) to positive (no alga contact)” would suffice.

Line 443: I see this is strongly correlative, not that this is bad, but high macroalgae cover may be one of many things at a site/region that are affecting microbial communities. I’d perhaps caution the “cause and effect” interpretation – as you eloquently put in the following paragraph and in the conclusion passage.

Line 499: or the abiotic conditions that favor microbial growth?

Line 517: no longer clades, say species/genera

Line 537: by site-level effect you really mean algal abundance, yes? Maybe saying the term instead of the effect would help in clarifying these passages.

Line 555: local and site-level algae? You mean intimate contact or macroalgae abundance?

Line 560: is it better to describe these groups as “putative” beneficial symbionts (as you do in Line 505)? I’m uncertain whether these groups are well linked to function/health the way this sentence suggests.

All figures in main text and supplement – Moorea needs okina as well

Fig. 1. I really like this figure!

Fig 2. Can you spell out interaction or define it in the legend?

Review form: Reviewer 2 (Craig Nelson)

Is the manuscript scientifically sound in its present form?

Yes

Are the interpretations and conclusions justified by the results?

Yes

Is the language acceptable?

Yes

Do you have any ethical concerns with this paper?

No

Have you any concerns about statistical analyses in this paper?

Yes

Recommendation?

Major revision is needed (please make suggestions in comments)

Comments to the Author(s)

Review attached because the system crashed on me (see Appendix A).

Decision letter (RSOS-210035.R0)

Dear Ms Briggs

The Editors assigned to your paper RSOS-210035 "Local vs. site-level effects of algae on coral microbial communities" have now received comments from reviewers and would like you to revise the paper in accordance with the reviewer comments and any comments from the Editors. Please note this decision does not guarantee eventual acceptance.

Please submit your revised manuscript and required files (see below) no later than 21 days from today's (ie 11-Mar-2021) date. Note: the ScholarOne system will 'lock' if submission of the revision is attempted 21 or more days after the deadline. If you do not think you will be able to meet this deadline please contact the editorial office immediately.

on behalf of Dr Nicole Hynson (Associate Editor) and Pete Smith (Subject Editor)
openscience@royalsociety.org

Associate Editor Comments to Author (Dr Nicole Hynson):

In line with the two expert Reviewer's comments I enjoyed reading this article and feel that it makes a nice contribution to our understanding of some of the factors that govern coral microbiomes. Similar to Reviewer 2, I would suggest the following: rather than reporting observed richness and Shannon diversity, consider reporting the first 3 Hill numbers (of which the first is observed richness, the second is Shannon entropy, and the third inverse Simpsons). Hill numbers have numerous advantages over other diversity indices commonly used for microbial studies. These are outlined in: Chao A, Gotelli NJ, Hsieh TC, Sander EL, Ma KH, Colwell RK, Ellison AM. 2014. Rarefaction and extrapolation with Hill numbers: a framework for sampling and estimation in species diversity studies. *Ecological Monographs* 84: 45–67. and can be easily calculated in the R package iNext.

Also, please be sure that you reference the relevant figures in your discussion as well as address the other, mostly minor, comments raised by both Reviewers.

I look forward to reading a revision of this MS.

Reviewer comments to Author:
Reviewer: 1
Comments to the Author(s)

Briggs et al. use 16S microbial data to test how coral microbiomes shift in response to changes in algal cover (across regional scales) and intimate contact between corals and algae or its absence. The authors present several well articulated hypotheses to explain their results and some nice conceptual figures as well. Ultimately the authors find that many metrics used to study coral microbiomes (alpha diversity, beta diversity) increase with macroalgae community %cover whereas the opposite is observed when corals are in close contact with algae AND algae % cover is increasing. These results suggest that changes in coral microbiomes are both a function of

intimate contact with algae, community macroalgae cover, and their interaction. This work, therefore, contributes to our understanding of coral microbiomes and the species-interactions (as well as community traits at varying scales) that lead to shifting microbial community composition.

Overall I found the paper well written, the experiment executed correctly and with appropriate statistical analyses (hats off to you! It is quite impressive). I would rate my comments as minor and seeking clarity overall.

Specifically, I find the terms "local effect and regional effect" to be confusing, as local to me implies a small spatial scale but not immediate proximity (i.e., touching). I don't want to discourage the authors from using the terms they want, but to me describing these terms, showing term fidelity, and re-emphasizing the "intimate contact" or proximity is key here. Similarly, the authors use both "site" and "regional" to describe larger spatial patterns across islands. I think again some term fidelity and using appropriate terms to best describe these effects when discussing them would be helpful. This need for clarity in terms and describing effects is, in my opinion, the most significant issue the manuscript faces, although this is easily addressed.

The results section is quite dense, but perhaps not overly so considering the amount of data to unpack. Perhaps some streamlining could be undertaken, but I appreciate if this cannot be done. I also appreciate the author's use of the supplement to provide further information on their study and its findings, which helped provide a lot of valuable information. In terms of data accessibility, a suggestion: it might be prudent to look into a 3rd party site to host the code/data (such as zenodo or Dryad, which I believe has a deal with RSOS). I see these files in the supplement, but a standalone repo would allow for the code (R project/Rmd/data) to be stored in a succinct directory making it easier for others to access and reproduce analyses.

Well done and I look forward to seeing this paper published.

Specific comments.

Line 12: spatial scales – for clarity, I think the authors mean “are these effects context dependent based on algal density”, which may not necessarily be spatially explicit but rather determined by a across a range of species-interaction. Perhaps rephrase to emphasize the 2 levels you are focusing on (1) are micorbial interactions varying across spatial scales, (2) and does this relate to macroalgae/turf density. I think you say this quite elegantly in Line 88.

Line 14: colon in stead of comma? (same at Line 90). Could flip to say Moorea and Mangareva, FP (country last – as you do in Line 115). Also, here and elsewhere you have an apostrophe in Moorea (Mo'orea), when it should be an okina (Mo'orea).

Line 20-21: what is “local algae”? density? In Line 21 perhaps rephrase: coral microbiomes are affected by both site-level algae density (in absence of direct contact) and proximity effects between coral and algae. As written the take home isn't punching through.

Line 26-28: “Coral reefs” (many different types of 'reefs'). Citation for the slip-to-slime, as it were?

Line 36/39: hereafter “microbiome?” To orient the reader to the term...

Line 54: is this to say the size of a site or the distance between sites? I also see the reference to “a local effect” in line 53 with the “local scales” in line 58 to be confusing. There is the local neighborhood (<1000 m²?) and the local neighbor (touching you in this case). Maybe there is a way to disentangle this semantically (i.e. how you discuss the “zone of interaction” in Line 61).

Line 59: citation to clarify this passage?

Line 63: can you again remind us what the distinction is between local and regional SITE scales? You describe this well in Line 344 heading as “local contact vs. site-level effects”. I suppose you

are also discussing within site and across a region here too (or is site = region? Seems Line 370 is stating this is the case...)? Some clarification would help.

Line 68-75: rephrase: (line 68) algae effects on coral microbiomes increase with regional density of algae? In general I find it hard to follow when “algae effect” is stated but what it is affecting is not – i.e., “algae effects on coral microbiomes...” (as in Line 73). Specifying the in-contact (<5cm?), local (<1000m²), region (>1000m²?) is important earlier to make this statement in Line 75 clear.

Line 108: last part of sentence here is not clear.

Line 119: I’d be cautious here, as these effects on islands are very often modified around the island. i.e., more human impact in Cook’s Bay with high sewage or groundwater discharge? This would be low-to-absent on south shore...

Line 138: missing a “)” after refs.

Line 159: use the micro symbol

Line 190: no italics on reference (and again at line 211, 246)

Line 198: citation as [44-46]? I’d make sure refs all formatted correctly at final version, such as here and Line 213 “[50,51]”

Line 223: “with” perhaps instead of “and” site as a random factor. New paragraph at the start of Line 223 sentence?

Line 253: “no-local scale”? or what does non-focal mean here?

Line 255: “0.05”

Line 309 and elsewhere: there should always be a zero before a decimal place. Please correct throughout.

Line 355: the “like it did” and “narrowly missed the cutoff” seems a bit too informal. Clarify the effect you are referencing and rephrase non-significant results as a trend?

Line 383: be consistent with or without spacing before “%”

Line 410: comma after e.g.

Line 412: algae density at regional scales? Coral microbiomes responding to this or something else (a different reef substrate)?

Line 441-442: I don’t see the phrasing here as making the point more clear. Can this be clarified? Perhaps even simply putting “switch from negative (in contact with algae) to positive (no alga contact)” would suffice.

Line 443: I see this is strongly correlative, not that this is bad, but high macroalgae cover may be one of many things at a site/region that are affecting microbial communities. I’d perhaps caution the “cause and effect” interpretation – as you eloquently put in the following paragraph and in the conclusion passage.

Line 499: or the abiotic conditions that favor microbial growth?

Line 517: no longer clades, say species/genera

Line 537: by site-level effect you really mean algal abundance, yes? Maybe saying the term instead of the effect would help in clarifying these passages.

Line 555: local and site-level algae? You mean intimate contact or macroalgae abundance?

Line 560: is it better to describe these groups as “putative” beneficial symbionts (as you do in Line 505)? I’m uncertain whether these groups are well linked to function/health the way this sentence suggests.

All figures in main text and supplement – Moorea needs okina as well

Fig. 1. I really like this figure!

Fig 2. Can you spell out interaction or define it in the legend?

Reviewer: 2

Comments to the Author(s)

Review attached because the system crashed on me.

===PREPARING YOUR MANUSCRIPT===

===PREPARING YOUR REVISION IN SCHOLARONE===

- An individual file of each figure (EPS or print-quality PDF preferred [either format should be produced directly from original creation package], or original software format).
- An editable file of each table (.doc, .docx, .xls, .xlsx, or .csv).
- An editable file of all figure and table captions.

- Any electronic supplementary material (ESM).
- If you are requesting a discretionary waiver for the article processing charge, the waiver form must be included at this step.
- If you are providing image files for potential cover images, please upload these at this step, and inform the editorial office you have done so. You must hold the copyright to any image provided.
- A copy of your point-by-point response to referees and Editors. This will expedite the preparation of your proof.

- Ensure that your data access statement meets the requirements at <https://royalsociety.org/journals/authors/author-guidelines/#data>. You should ensure that you cite the dataset in your reference list. If you have deposited data etc in the Dryad repository, please include both the 'For publication' link and 'For review' link at this stage.
- If you are requesting an article processing charge waiver, you must select the relevant waiver option (if requesting a discretionary waiver, the form should have been uploaded at Step 3 'File upload' above).
- If you have uploaded ESM files, please ensure you follow the guidance at <https://royalsociety.org/journals/authors/author-guidelines/#supplementary-material> to include a suitable title and informative caption. An example of appropriate titling and captioning may be found at https://figshare.com/articles/Table_S2_from_Is_there_a_trade-off_between_peak_performance_and_performance_breadth_across_temperatures_for_aerobic_scope_in_teleost_fishes_/3843624.

Author's Response to Decision Letter for (RSOS-210035.R0)

See Appendices B & C.

RSOS-210035.R1 (Revision)

Review form: Reviewer 1

Is the manuscript scientifically sound in its present form?

Yes

Are the interpretations and conclusions justified by the results?

Yes

Is the language acceptable?

Yes

Do you have any ethical concerns with this paper?

No

Have you any concerns about statistical analyses in this paper?

No

Recommendation?

Accept with minor revision (please list in comments)

Comments to the Author(s)

The authors have addressed all my comments and concerns and have made substantial improvements to the manuscript which have helped the both the structure and ease of interpretation (especially in the methods). My comments are quite minor and I apologize to the authors for taking so long to return my review. However, I wanted to give proper attention to the revision and the comments of reviewer 2 (who is much more of an expert than I am!). In short, I am happy to recommend the MS for publication.

PS - Excellent work on the conceptual figures. I think they really help communicate the findings and hypotheses.

My comments are on the clean version of the MS not the tracked changed version FYI.

As a small note it appears the "C" in °C in PCR protocol (see Line 31) is a different font than rest of paper.

Line 38 page 9: what is "1:30s"? Is this 90s? 1.5min?

Line 10 page 10: "v." for version

Line 4-22 page 12: this seems to be information for the results section. Perhaps you moved this around due to a request from the 2 reviewers, but it seems it would be best suited to go with results of PCA. If you used this as a way to test how your main effects should be structured/included in models then I see that as perhaps reasonable to include here. It is not clear to me how PCA was used (results are NMDS) so perhaps a little context as to why you used PCA would help. You do a nice job of describing the results of the PCA but less on the "why" which would be important here in the methods.

Line 38: page 17: there is a " _ " before the sentence starts

Line 45: underline the bold subheading like above sections? (and sans colon)

Review form: Reviewer 2 (Craig Nelson)

Is the manuscript scientifically sound in its present form?

Yes

Are the interpretations and conclusions justified by the results?

Yes

Is the language acceptable?

Yes

Do you have any ethical concerns with this paper?

No

Have you any concerns about statistical analyses in this paper?

No

Recommendation?

Accept as is

Comments to the Author(s)

The revised manuscript is excellent, the discussion well written, and the revisions appropriate (including the excellent explanations about the decision making process in addressing the sometimes disparate recommendations of the reviewers and editor). Both reviews were extensive and I think improved the manuscript significantly. I thought that the discussion on Figure 2 was well considered, and appreciate that the authors considered my suggestions thoughtfully and were clear when and why they disagreed.

I respect the disagreement with the authors on presenting means of non-normal data and whether this is valid. I don't disagree that calculating the mean of non-normal data can be useful, but I do disagree that comparing means or drawing quantile distributions like box-whisker diagrams is inappropriate unless the data are transformed to meet the assumptions of those approaches. Nonetheless, the authors are clearly well-versed in statistics and I applaud their determination to stick to their approaches and clearly articulate their reasoning, so I think this is all fine and good.

Fun note:

Although Reviewer 1 suggested "Also, here and elsewhere you have an apostrophe in Moorea (Mo'orea), when it should be an okina (Mo'orea)." and you responded (appropriately for the request) as follows "We were unaware of this distinction, thank you for pointing this out. We have replaced all apostrophes in Mo'orea with an okina." ...Reviewer 1 is actually sort of wrong, and you were sort of right. The grammar differs among polynesian languages, though all have some kind of consonant that represents a hardening of the palate (aka glottal stop), and in Tahitian the cognate of the hawaiian 'okina ('eta) is written as an apostrophe (whereas in Hawaiian the 'okina is written using the ASCII symbol for an "open quote")...so you are correct either way. Most Tahitian writing uses an apostrophe, so Mo'orea is most common.

Decision letter (RSOS-210035.R1)

Dear Ms Briggs,

On behalf of the Editors, we are pleased to inform you that your Manuscript RSOS-210035.R1 "Local vs. site-level effects of algae on coral microbial communities" has been accepted for publication in Royal Society Open Science subject to minor revision in accordance with the referees' reports. Please find the referees' comments along with any feedback from the Editors below my signature.

We invite you to respond to the comments and revise your manuscript. Below the referees' and Editors' comments (where applicable) we provide additional requirements. Final acceptance of

your manuscript is dependent on these requirements being met. We provide guidance below to help you prepare your revision.

Please submit your revised manuscript and required files (see below) no later than 7 days from today's (ie 16-Aug-2021) date. Note: the ScholarOne system will 'lock' if submission of the revision is attempted 7 or more days after the deadline. If you do not think you will be able to meet this deadline please contact the editorial office immediately.

on behalf of Dr Nicole Hynson (Associate Editor) and Pete Smith (Subject Editor)
openscience@royalsociety.org

Associate Editor Comments to Author (Dr Nicole Hynson):

The authors have done a sufficient job of revising their MS based on the Reviewer's comments and defending their rationale for not making some of the suggested changes. Reviewer 1 has a few minor comments to be addressed before publication and Reviewer 2 points out that the proper diacritical mark to honor the Tahitian written language would be an apostrophe in Mo'orea. I request that the authors make these minor changes.

Reviewer comments to Author:

Reviewer: 1
Comments to the Author(s)

The authors have addressed all my comments and concerns and have made substantial improvements to the manuscript which have helped the both the structure and ease of interpretation (especially in the methods). My comments are quite minor and I apologize to the authors for taking so long to return my review. However, I wanted to give proper attention to the revision and the comments of reviewer 2 (who is much more of an expert than I am!). In short, I am happy to recommend the MS for publication.

PS - Excellent work on the conceptual figures. I think they really help communicate the findings and hypotheses.

My comments are on the clean version of the MS not the tracked changed version FYI. As a small note it appears the "C" in °C in PCR protocol (see Line 31) is a different font than rest of paper.

Line 38 page 9: what is "1:30s"? Is this 90s? 1.5min?

Line 10 page 10: "v." for version

Line 4-22 page 12: this seems to be information for the results section. Perhaps you moved this around due to a request from the 2 reviewers, but it seems it would be best suited to go with results of PCA. If you used this as a way to test how your main effects should be structured/included in models then I see that as perhaps reasonable to include here. It is not clear to me how PCA was used (results are NMDS) so perhaps a little context as to why you used PCA would help. You do a nice job of describing the results of the PCA but less on the "why" which would be important here in the methods.

Line 38: page 17: there is a " " before the sentence starts

Line 45: underline the bold subheading like above sections? (and sans colon)

Reviewer: 2

Comments to the Author(s)

The revised manuscript is excellent, the discussion well written, and the revisions appropriate (including the excellent explanations about the decision making process in addressing the sometimes disparate recommendations of the reviewers and editor). Both reviews were extensive and I think improved the manuscript significantly. I thought that the discussion on Figure 2 was well considered, and appreciate that the authors considered my suggestions thoughtfully and were clear when and why they disagreed.

I respect the disagreement with the authors on presenting means of non-normal data and whether this is valid. I don't disagree that calculating the mean of non-normal data can be useful, but I do disagree that comparing means or drawing quantile distributions like box-whisker diagrams is inappropriate unless the data are transformed to meet the assumptions of those approaches. Nonetheless, the authors are clearly well-versed in statistics and I applaud their determination to stick to their approaches and clearly articulate their reasoning, so I think this is all fine and good.

Fun note:

Although Reviewer 1 suggested "Also, here and elsewhere you have an apostrophe in Moorea (Mo'orea), when it should be an okina (Mo'orea)." and you responded (appropriately for the request) as follows "We were unaware of this distinction, thank you for pointing this out. We have replaced all apostrophes in Mo'orea with an okina." ...Reviewer 1 is actually sort of wrong, and you were sort of right. The grammar differs among polynesian languages, though all have some kind of consonant that represents a hardening of the palate (aka glottal stop), and in Tahitian the cognate of the hawaiian 'okina ('eta) is written as an apostrophe (whereas in Hawaiian the 'okina is written using the ASCII symbol for an "open quote")...so you are correct either way. Most Tahitian writing uses an apostrophe, so Mo'orea is most common.

===PREPARING YOUR MANUSCRIPT===

===PREPARING YOUR REVISION IN SCHOLARONE===

- Ensure that your data access statement meets the requirements at <https://royalsociety.org/journals/authors/author-guidelines/#data>. You should ensure that you cite the dataset in your reference list. If you have deposited data etc in the Dryad repository, please only include the 'For publication' link at this stage. You should remove the 'For review' link.
- If you are requesting an article processing charge waiver, you must select the relevant waiver option (if requesting a discretionary waiver, the form should have been uploaded at Step 3 'File upload' above).
- If you have uploaded ESM files, please ensure you follow the guidance at <https://royalsociety.org/journals/authors/author-guidelines/#supplementary-material> to include a suitable title and informative caption. An example of appropriate titling and captioning may be found at https://figshare.com/articles/Table_S2_from_Is_there_a_trade-off_between_peak_performance_and_performance_breadth_across_temperatures_for_aerobic_scope_in_teleost_fishes_/3843624.

Author's Response to Decision Letter for (RSOS-210035.R1)

See Appendix D.

Decision letter (RSOS-210035.R2)

Dear Ms Briggs,

I am pleased to inform you that your manuscript entitled "Local vs. site-level effects of algae on coral microbial communities" is now accepted for publication in Royal Society Open Science.

You can expect to receive a proof of your article in the near future. Please contact the editorial office (openscience@royalsociety.org) and the production office (openscience_proofs@royalsociety.org) to let us know if you are likely to be away from e-mail contact -- if you are going to be away, please nominate a co-author (if available) to manage the proofing process, and ensure they are copied into your email to the journal. Due to rapid

publication and an extremely tight schedule, if comments are not received, your paper may experience a delay in publication.

on behalf of Dr Nicole Hynson (Associate Editor) and Pete Smith (Subject Editor)
openscience@royalsociety.org

Appendix A

Review of Amy Briggs, Anya Brown, Craig Osenberg for Proc. Open Science.
“Local vs. site-level effects of algae on coral microbial communities”

Briggs et al present a stellar study of interacting effects of local and regional algal influence on coral microbiomes.

Note to the editor: this paper is one I would nominate for a cover story or other highlight, as this is a very strong and widely interesting manuscript.

The conceptualization of this study is excellent and couched within a strong literature framework. The experimental design is solid and the statistical analyses are well done. I have some issues with a few aspects of the data analysis and presentation which I consider major revisions, but I expect that the authors will be able to make these changes and improve the manuscript to a level appropriate to the audience of this journal. Overall I think this is an excellent piece of work and I believe that the authors are underselling this.

Please note: I did not carefully review the discussion, because I believe it should be restructured and may need to be rewritten based off of some of my recommendations. I will review it during revision.

Major Points:

I advocate that Shannon Diversity be replaced or at least complemented with an evenness metric (Shannon Evenness or Pielou's J). Shannon Diversity and Richness are highly correlated in this study (which is common when richness gradients dominate the variation in Shannon Diversity Index) and are redundant, while evenness will at least be orthogonal.

Restructure so that all the models are presented together. It is reasonable to present univariate and multivariate graphics separately, but the model structures (lmer and adonis2) are nearly identical (although adonis2 does not allow the random effect of site) and can be presented to dovetail alpha and beta diversity. In this way the results and discussion can be separated mainly by model structure (there are 2 different frameworks, see my comment below) and thus by hypotheses being tested. In other words why not combine Table S3 and S4+S5 with Tables S6 and S7, respectively?

Along these same lines, when presenting models and results it is valuable to be explicit about the hypotheses being tested. This will allow the reader to better understand the modeling framework and goals.

Putting p values (and for adonis2/PERMANOVA also R²) on graphs for fixed effects in multivariate models will help a great deal. The supplement should only be a reference, not critical to interpreting the graphics. If you want to leave things like DF and F statistics in the supplement in tables that is fine, but the core statistics should be obvious on the graphics. I

would expect to see appropriate lmer results on Figure 2, S4, S5 and adonis2 results on Figures 3 and 4 and S6.

Given that you analyzed different species of coral, different species of macroalgae, and most likely different assemblages of turfing algae and different water masses, I would argue that you “know” that “island” will be a significant effect and you simply need to evaluate whether there is an island*cover interaction term. Since there is a universally significant interaction term, you can report these but then frame your subsequent analysis differently: Table S3 and Table S6 effectively argue that Tables S4-S5 and S7-S8 should be run separately for each island, with Table S4 and the “coral parts” of Tables S7-S8 being run to include the interaction of contact and cover.

Along these lines, it is worth noting that Figure 4 implies a PERMANOVA (adonis2) model that doesn't exist as far as I can tell from Table S7, but it should given your hypotheses in Figure S1: Figure 4 implies a model like “For each Island BCdistance~cover*contact” where you report R2 and p on each panel for all three terms. I advocate you run this model and provide those statistics to support the visualization provided in Figure 4.

Overall, it is my feeling that the manuscript will read much smoother if you consider the above and follow this structure: I think the figures are fine as is (with some tweaks as noted and just add the stats as I suggested):

1) alpha diversity (richness and evenness by lmer) and beta diversity (dispersion lmer and distance adonis2) ~ island*cover : results shown in Figures 3 and S4 with statistics on the figures for each of the three fixed effects (p for lmer, p and R2 for adonis2).

2) Only Corals, separated by island, alpha diversity (richness and evenness by lmer) and beta diversity (dispersion lmer and distance adonis2) ~ cover*contact : results shown in Figures 2 and 4 with stats on the figures for each of the three fixed effects (p for lmer, p and R2 for adonis2) - Note that crucially this demands that the ordinations in Figure 4 be done separately for each island, which I currently think they are not (which is inappropriate regardless).

Figure 4 and L359-367: This is an important test of some of your hypotheses (particularly Figure S1) and the discussion should be more quantitative and less qualitative. NMDS are a visualization tool, not statistics, so I would endeavor to support these statements with statistical data before asking the reader to see the pattern in an NMDS. You can run pairwise.adonis which might allow you to test if the distance between treatments nocontact and algae is smaller or larger than the distance between turfcontact or macroalgaecontact and algae at your three different macroalgae cover levels (which may need to be condensed?). I also advocate considering a regression approach, where the bray-curtis distance between coral and algae (y) is related to %cover separately for each of the three contact types separately for each island. You can do this by decomposing your distance matrix and annotating all pairwise bray-curtis distances between these categories on each island as one of the three types (nocontact-algae, turfcontact-algae, macrocontact-algae).

Finally, for your thought and consideration, I find Figure 2 compelling and want you to think about some aspects and ways to bring this concept to the forefront, perhaps even with a new conceptual framework for how algal impacts work (and even how local vs. regional stressors work). I accept that you may not agree with or be able to find these patterns, but I do think you should consider them because your data appear stronger and more interesting than you give them credit for.

1) I think it would be worth testing if the Turf vs. Macroalgal contrast is significant (doesn't look like it) and then you can report that and on the figure just contrast coral with algaecontact. Will make a much clearer story with just two regression lines instead of three.

2) It seems that you may have a pattern that isn't in figure 1, namely local-regional compensation, where high enough regional effects (high enough algal diversity) overcome a lack of local effects and position the response variables at the "start" of local impacts (ie direct algal contact with minimal regional effects). In this sense, you might envision an additive continuous "algal impact" score, from nocontact-lowcover to nocontact-highcover to contact-lowcover to contact-highcover, which based on your data I visualize may have some kind of midpoint maximum, where richness, diversity and dispersion all increase and then decline.

3) It also suggests that examining gradients of community structure (possibly island-specific NMDS axes - see my comments for Figure 4?) along these same axes would be useful. Another gradient of community change you could examine would be something like mean bray-curtis distance from a "control" (nocontact-lowcover) - again see my comments for Figure 4.

Minor Revisions:

Title: why not change "site-level" to "regional"...this both sounds better and better reflects your model structure (which used site as a random effect)?

References: reference 6 and 22 are the same.

L54-56: I would add that Kelly et al 2014 also showed how benthic cover gradients in algal dominance related to the taxonomic composition of the coral reef microbiome...while not specifically an examination of regional effects on the CORAL microbiome this study did set the stage for benthic cover influencing reef microbiomes orthogonal to biogeochemical and physical parameters.

L117-119: I would also argue that it is not just the population of Mo'orea but also the influence of Tahiti (ie daily traffic)...not sure how best to phrase or quantify but worth pointing out to the uninitiated so it is apparent that Moorea is not an isolated island of 17k people but a satellite of the most populous island in the south pacific.

L175: What is meant in this sentence? What is being “determined”?

L188: You never indicate whether you worked primarily with relative abundances, particularly with corncob and the permanova. This is important.

L213: I think you mean dispersion, not beta diversity (your permanovas are also analyzing beta diversity)

L288: I am skeptical of the utility of the samples with less than several hundred reads, though I could be convinced. One way to explore this is a graphic relating read count to richness...in theory there will be some asymptote above which read count arguably does not appreciably influence richness...I advocate applying a minimum threshold there and excluding from all analyses samples below that number of reads.

L289: Please provide a supplemental reads/sample distribution was so we have some idea of how much variation and general statistics (mean, median, minimum, maximum). Note that means can't be calculated on non-normal data, so more than likely you need to log₁₀ the reads/sample to get a good mean (or use a geometric mean).

L290: In part to solve the problem in L288 it can be useful (and I advise) to apply a threshold of prevalence and maximum relative abundance below which ASVs are excluded from alpha diversity analyses. Please consider, especially if the graph I suggest on L288 makes you feel like you will lose a lot of samples you might get a better graph if you eliminate ASVs that are only in a few samples (say <3) or never have a relative abundance greater than X in more than 1 sample (X might be 0.1%)...this can smooth out the effects of imbalanced sampling effort (reads/sample).

L293: Clarify how you “rarefied”. Presumably this was not by true rarefaction (a modeling approach to predict richness at a lower sampling effort == reads/sample) but by random subsampling at a standardized reads/sample effort: give that number and be clear about the approach. You declare 74 reads in the legend for Figure S4...this is very low (See my comment on L288) and it would be better if it were a higher number. I think most microbial ecologists like 10,000, but I am comfortable with a few hundred or ideally 1000. Coral microbiome papers often have low reads because corals are a pain to amplify...

Figure 1: I would label the Y as “microbiome metric” rather than “microbiome response” as this seems like you are saying that the magnitude of the response, rather than the magnitude of a variable (say richness) changes with algal density.

Figure 2: I don't understand why you would do this: “Significant main effects are not shown if the interaction is significant”. I suspect my recommendation that you put the Imer/adonis2 stats on the figures will eliminate the need for this comment.

L382-383: See my comments on Figs S7-S8 about medians on untransformed data.

Figure 5 and 6: I would label the x-axis to clarify the differential test. By this I mean your figure should be interpretable without reading the legend. You state in the legend for Figure 5 “Positive numbers indicate taxa that increased with site-level macroalgal cover, negative numbers indicate taxa that declined with macroalgal cover” so just label one side of the x-axis “increased with macroalgal cover” and the other side “declined with macroalgal cover”. On Figure 6 label one side “increased with algal contact” etc...

Supplement - why is PCR protocol and filtering bioinformatics not in main manuscript? They are quite short, and the filtering bioinformatics are even redundant with the text in L179-181

Supplement - Why in PCR protocol is there “10 s at 78 °C” - unusual, inexplicable, and should be justified.

L234-235: Why are you using chi-square tests instead of lmer test or emmeans to evaluate p-values and contrasts? I am familiar with the car package but just haven't used it and don't know why it would apply a chi-squared test. I'd appreciate your advice here...I am just not familiar with the approach.

L232: I am not fond of this approach because it doesn't really match to testing the hypotheses driving your study. I advocate you structure specific models to test specific hypotheses rather than just whittling the largest models based on stepwise approaches...those are designed for a different question such as “what suite of variables best describe variation” not “how do variables A, B, C interact to predict Y”...the latter fits your deliberate study design much better.

Figure S7 and S8 should be presented after transform to normal distribution: you can't put normal quantile box plots on non-normal data, and relative abundances are rarely normal (and they certainly aren't here). Moreover a reader can't really understand these distributions when most of the families hover near zero. Statistically I recommend using the angular transform ($\text{asin}(\sqrt{x})$) but graphically \log_{10} is much easier to visualize: for visualization with \log_{10} you can convert all zeros to a minimum based on your subsampling/rarefaction level (ie a limit of detection - say 0.01% or something). But if/when you do statistical tests (which I don't think you are doing on these?) it is best to use another transformation to approximate normality such as angular transform. Note that the “median” criteria for inclusion should likewise be calculated on transformed data.

Figure S8c I recommend “Clade_I” be annotated with what group it actually is (Probably “SAR11 Clade_I” based on figs 5-6).

Figures 5,6,S7,S8: In general I recommend organizing the order of families on Y axis by Phylum/Class and annotating the order/classes so the reader has some frame of reference (you add the order on Figs 5-6 but not S7-S8). You might also just double-check that “Cyanobiaceae” is indeed the family for what I assume is *Synechococcus*?

Signed: Craig Nelson (in case you want to reach out for clarification or advice)

Appendix B

Responses to reviewers for manuscript “Local vs. site-level effects of algae on coral microbial communities” by

Amy A. Briggs, Anya L. Brown, Craig W. Osenberg

Key: Editor and Reviewer comments in black text. Author responses in blue text.

Associate Editor Comments to Author (Dr Nicole Hynson):

In line with the two expert Reviewer's comments I enjoyed reading this article and feel that it makes a nice contribution to our understanding of some of the factors that govern coral microbiomes. Similar to Reviewer 2, I would suggest the following: rather than reporting observed richness and Shannon diversity, consider reporting the first 3 Hill numbers (of which the first is observed richness, the second is Shannon entropy, and the third inverse Simpsons). Hill numbers have numerous advantages over other diversity indices commonly used for microbial studies. These are outlined in: Chao A, Gotelli NJ, Hsieh TC, Sander EL, Ma KH, Colwell RK, Ellison AM. 2014. Rarefaction and extrapolation with Hill numbers: a framework for sampling and estimation in species diversity studies. *Ecological Monographs* 84: 45–67. and can be easily calculated in the R package iNext.

Also, please be sure that you reference the relevant figures in your discussion as well as address the other, mostly minor, comments raised by both Reviewers.

I look forward to reading a revision of this MS.

We thank the editor and reviewers for their detailed and thoughtful comments. As the editor has suggested, we have graphically reported our alpha diversity results as Hill numbers, rather than diversity indices. However, due to statistical characteristics of the data in the Hill number format, we have had to use log-transformed versions of this data for our statistical analyses (which converts Hill number order 1 back to Shannon entropy). We only reported the Hill numbers of order 1 and 2 (Shannon and Simpson diversity), but did not report richness (Hill number order 0), as we believe our sampling depth was too low to accurately estimate richness for a very small subset of samples (which is why we previously subsampled to an even read depth for our richness analyses). Additionally, we used two, rather than three Hill numbers to appease the concerns of Reviewer 2 that multiple alpha diversity results would be correlated. We thank the editor for information about the iNEXT package. However, the extrapolated diversity estimates using iNEXT did not demonstrably differ from our observed values, even for low-read depth samples, so we used our observed data.

We have addressed the comments of the reviewers by adding two graphs to Fig. 4, adding many additional annotations and statistical output to our graphs, re-doing some statistical analyses by island, and re-organizing our results section. Below we provide specific responses to the reviewers.

Reviewer comments to Author:

Reviewer: 1

Comments to the Author(s)

Briggs et al. use 16S microbial data to test how coral microbiomes shift in response to changes in algal cover (across regional scales) and intimate contact between corals and algae or its absence. The authors present several well articulated hypotheses to explain their results and some nice conceptual figures as well. Ultimately the authors find that many metrics used to study coral microbiomes (alpha diversity, beta diversity) increase with macroalgae community % cover whereas the opposite is observed when corals are in close contact with algae AND algae % cover is increasing. These results suggest that changes in coral microbiomes are both a function of intimate contact with algae, community macroalgae cover, and their interaction. This work, therefore, contributes to our understanding of coral microbiomes and the species-interactions (as well as community traits at varying scales) that lead to shifting microbial community composition.

Overall I found the paper well written, the experiment executed correctly and with appropriate statistical analyses (hats off to you! It is quite impressive). I would rate my comments as minor and seeking clarity overall.

Specifically, I find the terms "local effect and regional effect" to be confusing, as local to me implies a small spatial scale but not immediate proximity (i.e., touching). I don't want to discourage the authors from using the terms they want, but to me describing these terms, showing term fidelity, and re-emphasizing the "intimate contact" or proximity is key here. Similarly, the authors use both "site" and "regional" to describe larger spatial patterns across islands. I think again some term fidelity and using appropriate terms to best describe these effects when discussing them would be helpful. This need for clarity in terms and describing effects is, in my opinion, the most significant issue the manuscript faces, although this is easily addressed.

We agree that "local" and "regional" are relative terms that can refer to different specific scales in different studies. We have tried to be more consistent in using "algal contact" and "site-level macroalgal cover" in our methods, results, and discussion, but we have kept local vs. regional in our introduction and some areas of our discussion when we are discussing these spatial scales in a more general or theoretical sense.

The results section is quite dense, but perhaps not overly so considering the amount of data to unpack. Perhaps some streamlining could be undertaken, but I appreciate if this cannot be done. I also appreciate the author's use of the supplement to provide further information on their study and its findings, which helped provide a lot of valuable information.

We have attempted to streamline the section regarding the responses of specific microbial families in the coral samples to algal contact and macroalgal cover.

In terms of data accessibility, a suggestion: it might be prudent to look into a 3rd party site to host the code/data (such as zenodo or Dryad, which I believe has a deal with RSOS). I see these files in the supplement, but a standalone repo would allow for the code (R project/Rmd/data) to be stored in a succinct directory making it easier for others to access and reproduce analyses.

Now that we have finalized our analyses based on reviewers' comments, we have moved our data and code to Dryad.

Well done and I look forward to seeing this paper published.

(Thank you, we appreciate the kind feedback!)

Specific comments.

Line 12: spatial scales – for clarity, I think the authors mean “are these effects context dependent based on algal density”, which may not necessarily be spatially explicit but rather determined by a across a range of species-interaction. Perhaps rephrase to emphasize the 2 levels you are focusing on (1) are micorbial interactions varying across spatial scales, (2) and does this relate to macroalgae/turf density. I think you say this quite elegantly in Line 88.

We have a more specific meaning than the reviewer suggests, as density can be evaluated at multiple spatial scales (e.g., algae density in a 1 x 1 m quadrat or algal density in a 100 x 100 m site). In the sentences that the reviewer references, “microbes mediate negative effects of algae on corals when corals are in contact with algae. However, it is unknown whether these effects extend to larger spatial scales, such as at sites with high algal densities,” we think that it is clear that we are contrasting effects of algae on corals at smaller spatial scales (contact) with effects at larger spatial scales, and we also provide specific examples of what those two scales are.

Line 14: colon in stead of comma? (same at Line 90). Could flip to say Moorea and Mangareva, FP (country last—as you do in Line 115).

This would appear to be a matter of style and we would prefer to retain our original usage: “two islands in French Polynesia, Moorea and Mangareva.”

Also, here and elsewhere you have an apostrophe in Moorea (Mo'orea), when it should be an okina (Mo'orea).

We were unaware of this distinction, thank you for pointing this out. We have replaced all apostrophes in Mo'orea with an okina.

Line 20-21: what is “local algae”? density? In Line 21 perhaps rephrase: coral microbiomes are affected by both site-level algae density (in absence of direct contact) and proximity effects between coral and algae. As written the take home isn't punching through.

“Proximity effects” implies evaluation of the effects of algae along a continuous distance scale, which we didn’t do. We are also constrained by the word limit for the abstract. We have therefore retained our original language which is more succinct.

Line 26-28: “Coral reefs” (many different types of ‘reefs’). Citation for the slip-to-slime, as it were?

We added the word “coral” to be more specific, and have added additional citations related to the disturbances threatening coral reefs, and factors leading to algal proliferation.

Line 36/39: hereafter “microbiome?” To orient the reader to the term...

We changed the text to clearly define the term microbiome here, and have tried to use it more throughout the paper.

Line 54: is this to say the size of a site or the distance between sites? I also see the reference to “a local effect” in line 53 with the “local scales” in line 58 to be confusing. There is the local neighborhood (<1000 m²?) and the local neighbor (touching you in this case). Maybe there is a way to disentangle this semantically (i.e. how you discuss the “zone of interaction” in Line 61).
Line 59: citation to clarify this passage?

Old line 54: This is a rough estimate of the area (hence the units, m²) characterized by the sampling design at each site in the cited studies. Area estimates are based on the transect length, the number of transects, and the distance between transects at each site. Distance between sites does not seem to be a good way to characterize a site’s spatial extent, as there can be a large amount of unsampled area between sites (some of which may be open ocean, or not analogous habitat).

A local effect is an effect that occurs over a local spatial scale. Local is defined as close proximity or touching. We have added “(< 1 m²)” to “experimental studies that have demonstrated effects of algae on coral microbiomes all focus on corals that were in close proximity (< 1 m²) or in contact with algae: i.e., a “local” effect.”

Old Lines 58-59: We have changed “...dilution and benthic boundary layers might prevent effects at larger spatial scales.” to, “...effects at larger (*i.e., regional*) spatial scales.” However, this is a hypothetical statement, so we cannot provide a citation to it, as requested.

Line 63: can you again remind us what the distinction is between local and regional SITE scales? You describe this well in Line 344 heading as “local contact vs. site-level effects”. I suppose you are also discussing within site and across a region here too (or is site = region? Seems Line 370 is stating this is the case...)? Some clarification would help.

At the beginning of this paragraph, we link our definition of a regional scale to sites: “Although there are no studies that document effects of algae on corals at larger “regional” scales (e.g., among sites that are each 1,000-10,000 m²).” Our intent is to contrast the absence of studies at

these scales, while there are studies conducted at the local scale, which we mentioned just prior to this sentence: "...corals that were in close proximity or in contact with algae: i.e., a "local" effect..."

Line 68-75: rephrase: (line 68) algae effects on coral microbiomes increase with regional density of algae? In general I find it hard to follow when "algae effect" is stated but what it is affecting is not—ie., "algae effects on coral microbiomes..." (as in Line 73). Specifying the in-contact (<5cm?), local (<1000m²), region (>1000m²?) is important earlier to make this statement in Line 75 clear.

We added ... "on the coral microbiome..." to the next sentence to emphasize what the algae are influencing. We use "local" and "regional" here rather than "in contact" and "site-level" because we are discussing these interactions in the abstract sense, and conceivably, local effects could arise from direct contact with, or close proximity to, algae (e.g., < 1 m²). We are not defining the spatial limits of these effects with this study. Instead, we investigate effects at two contrasting scales, so we want to use terms for the scales to be more general in our introduction. We are, however, more specific in our methods and results.

Line 108: last part of sentence here is not clear.

We have changed the sentence "Contrary to previous hypotheses, we found that algae affected multiple aspects of coral microbiomes at both local and regional scales, and that these different scales modified one another's effects." to "...we found that algae affected multiple aspects of coral microbiomes at both local and regional scales, *and that the effects of algae at one scale modified its effects at the other scale.*"

Line 119: I'd be cautious here, as these effects on islands are very often modified around the island. i.e., more human impact in Cook's Bay with high sewage or groundwater discharge? This would be low-to-absent on south shore...

We have added text "at least some of its reefs" to the statement about differences in anthropogenic influence.

Line 138: missing a ")" after refs. Changes made.

Line 159: use the micro symbol. Changes made.

Line 190: no italics on reference (and again at line 211, 246). Changes made.

Line 198: citation as [44-46]? I'd make sure refs all formatted correctly at final version, such as here and Line 213 "[50,51]"

We have checked our citation formats.

Line 223: “with” perhaps instead of “and” site as a random factor. Changes made.

New paragraph at the start of Line 223 sentence?

We started a new paragraph for the discussion of why we used macroalgal cover as a predictor.

Line 253: “no-local scale”? or what does non-focal mean here?

Non-focal means the scale whose effect we are not testing in this analysis. In these analyses, to evaluate the effect at one scale, we control for the effect at the other (non-focal) scale. We have tried to clarify this in the text.

Line 255: “0.05” Changes made.

Line 309 and elsewhere: there should always be a zero before a decimal place. Please correct throughout. Changes made.

Line 355: the “like it did” and “narrowly missed the cutoff” seems a bit too informal. Clarify the effect you are referencing and rephrase non-significant results as a trend?

Because we have re-run the PERMANOVA separately by island, our results and the associated text have changed.

Line 383: be consistent with or without spacing before “%”. We have inserted a space between a number and the % symbol throughout manuscript.

Line 410: comma after e.g. Changes made.

Line 412: algae density at regional scales? Coral microbiomes responding to this or something else (a different reef substrate)?

We have changed the words “due to” to “with” to emphasize this is a correlation. Additionally, we mention in the discussion that other site-level factors could be driving the association between site-level macroalgal cover and changes in the coral microbiome.

Line 441-442: I don't see the phrasing here as making the point more clear. Can this be clarified? Perhaps even simply putting “switch from negative (in contact with algae) to positive (no alga contact)” would suffice.

The suggested rephrasing is not accurate, so we have retained the original wording, but replaced “regional” with “site-level” to address the reviewer's other concerns.

Line 443: I see this is strongly correlative, not that this is bad, but high macroalgae cover may be one of many things at a site/region that are affecting microbial communities. I'd perhaps caution

the “cause and effect” interpretation—as you eloquently put in the following paragraph and in the conclusion passage.

We have changed the word “were” to “could be” to clarify that the hypothetical nature of our interpretation in this sentence.

Line 499: or the abiotic conditions that favor microbial growth?

We have added the text, “change abiotic conditions and/or…” to include changes other than increases in algal by-products.

Line 517: no longer clades, say species/genera
Changed to species

Line 537: by site-level effect you really mean algal abundance, yes? Maybe saying the term instead of the effect would help in clarifying these passages.

Here, “site-level effect of macroalgae” means the effect of macroalgal cover averaged across the site-scale. We have changed the text to reflect this.

Line 555: local and site-level algae? You mean intimate contact or macroalgae abundance?

Yes. We changed the text to “algal contact and site-level macroalgal cover” to be more explicit.

Line 560: is it better to describe these groups as “putative” beneficial symbionts (as you do in Line 505)? I’m uncertain whether these groups are well linked to function/health the way this sentence suggests.

We added the word “putative.”

All figures in main text and supplement—Moorea needs okina as well. Changes made.

Fig. 1. I really like this figure! Thank you.

Fig 2. Can you spell out interaction or define it in the legend?

We have added definitions of our abbreviation for macroalgal cover (macro. cover) to the caption and changed alg. contact to algal contact.

Reviewer: 2

Comments to the Author(s)

Review attached because the system crashed on me.

Review of Amy Briggs, Anya Brown, Craig Osenberg for Proc. Open Science. “Local vs. site-level effects of algae on coral microbial communities”

Briggs et al present a stellar study of interacting effects of local and regional algal influence on coral microbiomes.

Note to the editor: this paper is one I would nominate for a cover story or other highlight, as this is a very strong and widely interesting manuscript.

The conceptualization of this study is excellent and couched within a strong literature framework. The experimental design is solid and the statistical analyses are well done. I have some issues with a few aspects of the data analysis and presentation which I consider major revisions, but I expect that the authors will be able to make these changes and improve the manuscript to a level appropriate to the audience of this journal. Overall I think this is an excellent piece of work and I believe that the authors are underselling this.

We thank the reviewer for his kind words and thoughtful review.

Please note: I did not carefully review the discussion, because I believe it should be restructured and may need to be rewritten based off of some of my recommendations. I will review it during revision.

Major Points:

I advocate that Shannon Diversity be replaced or at least complemented with an evenness metric (Shannon Evenness or Pielou’s J). Shannon Diversity and Richness are highly correlated in this study (which is common when richness gradients dominate the variation in Shannon Diversity Index) and are redundant, while evenness will at least be orthogonal.

The editor and both reviewers had contrasting suggestions about how to handle these analyses. We have attempted to address their collective comments in spirit (if not exactly as proposed), given the disparate advice. We believe Shannon diversity (Hill number order 1) and Simpson diversity (Hill number order 2) are better estimates of alpha diversity than unrarefied richness (Hill number order 0) in our study, due to the low read depth for a portion of our samples. Shannon Diversity incorporates evenness and can accurately estimate alpha diversity even at low sampling intensity. It is also robust to zeroes and skewed relative abundances of taxa (Chao and Shen 2003, Lemos et al. 2013).

Since evenness is correlated with both Shannon and Simpson diversity, we have decided not to separately evaluate an evenness metric. Describing the change in the effective number of taxa for each Hill number allows us to discuss how dominant taxa contribute to diversity, which will address some of Reviewer 2's request that we explicitly consider evenness. A decline in the Hill number with increasing orders of q indicates that the community is uneven.

Chao, A., Chiu, C.- H., & Jost, L. (2014). Unifying species diversity, phylogenetic diversity, functional diversity, and related similarity and differentiation measures through Hill numbers. *Annual Review of Ecology Evolution and Systematics*, 45, 297–324.

Chao, A.; Shen, T. 2003. Nonparametric estimation of Shannon's index of diversity when there are unseen species in sample. *Environ. Ecol. Stat.*, 10, 429-433.

Lemos LN, Fulthorpe RR, Triplett EW, Roesch LFW. 2011 Rethinking microbial diversity analysis in the high throughput sequencing era. *J. Microbiol. Methods* 86, 42–51. (doi:10.1016/j.mimet.2011.03.014)

Restructure so that all the models are presented together. It is reasonable to present univariate and multivariate graphics separately, but the model structures (lmer and adonis2) are nearly identical (although adonis2 does not allow the random effect of site) and can be presented to dovetail alpha and beta diversity. In this way the results and discussion can be separated mainly by model structure (there are 2 different frameworks, see my comment below) and thus by hypotheses being tested. In other words why not combine Table S3 and S4+S5 with Tables S6 and S7, respectively?

Along these same lines, when presenting models and results it is valuable to be explicit about the hypotheses being tested. This will allow the reader to better understand the modeling framework and goals.

We have reorganized our results section into a section comparing microbiome responses among substrate types and islands, and a section evaluating the effects of local vs. site-level algae on microbiome responses. We have also tried to be more specific about our hypotheses in the methods section when we describe our modeling approach.

Putting p values (and for adonis2/PERMANOVA also R^2) on graphs for fixed effects in multivariate models will help a great deal. The supplement should only be a reference, not critical to interpreting the graphics. If you want to leave things like DF and F statistics in the supplement in tables that is fine, but the core statistics should be obvious on the graphics. I would expect to see appropriate lmer results on Figure 2, S4, S5 and adonis2 results on Figures 3 and 4 and S6.

We appreciate that it is easier to have statistical results indicated on the graphs, at least when the figure and the statistics are simple. We have done that in figures in which we think the statistics will not detract from the readability of the graphs (e.g., Fig. 2, 3) However, adding too much text to a plot detracts from the ability to read it, and we were unable to add all the requested statistics to some of our figures using a legible font size (e.g., adding PERMANOVA results to Fig. 4).

Since the statistical results are provided in the main text, and we indicate in figure captions where to look at in the supplement to find statistical summaries, we believe we have done due diligence to help the reader find the results. We also do not feel that it is necessary to add results to supplemental graphs when the table summarizing those results is on the next page.

Given that you analyzed different species of coral, different species of macroalgae, and most likely different assemblages of turfing algae and different water masses, I would argue that you “know” that “island” will be a significant effect and you simply need to evaluate whether there is an island*cover interaction term. Since there is a universally significant interaction term, you can report these but then frame your subsequent analysis differently: Table S3 and Table S6 effectively argue that Tables S4-S5 and S7-S8 should be run separately for each island, with Table S4 and the “coral parts” of Tables S7-S8 being run to include the interaction of contact and cover.

The specific identities of taxa contributing to diversity metrics at each island did not affect the diversity calculations (e.g., the corals at each island could have completely different taxa in their microbiomes but still have similar richness or beta dispersion). Additionally, we did not necessarily know that diversity would be demonstrably different by island, so we thought it was a predictor worth testing. When we tested models containing an island x macroalgal cover interaction, this interaction was *not* significant for any of our diversity metrics in Fig 2 (corals) or for our algae and water samples. Therefore, we did not do separate tests by island for the alpha diversity and beta dispersion data, which would have reduced the power of using our whole dataset to estimate our parameters, and inflated the probability of Type I errors by increasing the number of statistical tests that we performed.

The cornucopia analyses were separated by island because these models dealt with the relative abundances of specific microbial taxa, which were not always found in both coral species. However, since community composition deals with specific microbial taxa, we have now run separate analyses for each island to test the effects of local algal contact and macroalgal cover.

Along these lines, it is worth noting that Figure 4 implies a PERMANOVA (adonis2) model that doesn't exist as far as I can tell from Table S7, but it should given your hypotheses in Figure S1: Figure 4 implies a model like “For each Island BCdistance~cover*contact” where you report R² and p on each panel for all three terms. I advocate you run this model and provide those statistics to support the visualization provided in Figure 4.

We did perform a PERMANOVA on the coral samples testing if Bray Curtis dissimilarity of samples was different with site-level macroalgal cover and algal contact. These results were summarized in the text, and we indicated in the figure caption the supplemental table that summarized the PERMANOVA results. To add the full PERMANOVA results to the plot, the text would need to be too small to be readable. However, we have added the p-values. We also added two panels to Fig. 4 that show the position of coral and macroalgae samples along the primary axis separating the coral and algae samples, NMDS1, vs. site-level macroalgal cover. These panels show the pattern we described in the text, and we have added the results for a regression fit to the data for each island testing the effects of macroalgal cover and algal contact on NMDS1.

Overall, it is my feeling that the manuscript will read much smoother if you consider the above and follow this structure: I think the figures are fine as is (with some tweaks as noted and just add the stats as I suggested):

1) alpha diversity (richness and evenness by lmer) and beta diversity (dispersion lmer and distance adonis2) ~ island*cover : results shown in Figures 3 and S4 with statistics on the figures for each of the three fixed effects (p for lmer, p and R2 for adonis2).

2) Only Corals, separated by island, alpha diversity (richness and evenness by lmer) and beta diversity (dispersion lmer and distance adonis2) ~ cover*contact : results shown in Figures 2 and 4 with stats on the figures for each of the three fixed effects (p for lmer, p and R2 for adonis2) - Note that crucially this demands that the ordinations in Figure 4 be done separately for each island, which I currently think they are not (which is inappropriate regardless).

We did ordinations separately by island in Fig 4. We used Shannon and Simpson diversity rather than richness and evenness (justification in our response to the first major point in the review). We did not run separate statistical tests by island for diversity (justification in our response to the earlier comment, "Given that you analyzed different species of coral,...").

Figure 4 and L359-367: This is an important test of some of your hypotheses (particularly Figure S1) and the discussion should be more quantitative and less qualitative. NMDS are a visualization tool, not statistics, so I would endeavor to support these statements with statistical data before asking the reader to see the pattern in an NMDS. You can run pairwise.adonis which might allow you to test if the distance between treatments nocontact and algae is smaller or larger than the distance between turfcontact or macroalgaecontact and algae at your three different macroalgae cover levels (which may need to be condensed?).

Pairwise tests would allow us to test if our groups (either coral local contact groups or coral local contact groups vs. macroalgae) are significantly different and if they respond to macroalgae cover differently, however, we still need to use a graphical display to determine if the dissimilarities are larger for one group vs. another.

I also advocate considering a regression approach, where the bray-curtis distance between coral and algae (y) is related to %cover separately for each of the three contact types separately for each island. You can do this by decomposing your distance matrix and annotating all pairwise bray-curtis distances between these categories on each island as one of the three types (nocontact-algae, turfcontact-algae, macrocontact-algae).

PERMANOVA for the dissimilarities between coral vs. algae samples was not possible, because PERMANOVA requires a square distance matrix, meaning coral vs. coral and algae vs. algae dissimilarities would also need to be included. Raw Bray-Curtis dissimilarities were not used in a regression because they were not independent (1000s of combinations of coral vs. algae samples for each island). The best (albeit, statistically imperfect) approach we could take to test the dissimilarity between coral vs. macroalgae microbiomes was to fit linear regression models containing algal contact, site-level macroalgal cover, and their interaction, to the NMDS1 response for each coral and macroalgae sample at each island. However, NMDS ordination

distances are not independent, so results should be interpreted cautiously. We have added this last approach (and associated caveat) to our results and Fig. 4.

Finally, for your thought and consideration, I find Figure 2 compelling and want you to think about some aspects and ways to bring this concept to the forefront, perhaps even with a new conceptual framework for how algal impacts work (and even how local vs. regional stressors work). I accept that you may not agree with or be able to find these patterns, but I do think you should consider them because your data appear stronger and more interesting than you give them credit for.

We describe a new conceptual framework that could explain the patterns in Fig 2 in the discussion (with the colonization/differential growth tradeoffs). We believe it is most appropriate to discuss this framework in there, rather than the introduction, as we do not know of any evidence published prior to our study to suggest a greater-than-antagonistic interaction between local and regional algal effects. We also did not measure some of the variables (like microbial density) that would be necessary to test the new mechanistic framework that we proposed in our discussion (e.g., effects of algae on microbial densities influencing colonization rates).

1) I think it would be worth testing if the Turf vs. Macroalgal contrast is significant (doesn't look like it) and then you can report that and on the figure just contrast coral with algaecontact. Will make a much clearer story with just two regression lines instead of three.

There are some differences in the responses of beta dispersion and microbial community composition to contact with macroalgae vs. turf algae (e.g., slope differences in response of beta dispersion to macroalgal cover for the two contact groups). Additionally, we have a priori reasons to believe the different types of algae could have different effects. Therefore, we chose not to aggregate the data.

2) It seems that you may have a pattern that isn't in figure 1, namely local-regional compensation, where high enough regional effects (high enough algal diversity) overcome a lack of local effects and position the response variables at the "start" of local impacts (ie direct algal contact with minimal regional effects). In this sense, you might envision an additive continuous "algal impact" score, from nocontact-lowcover to nocontact-highcover to contact-lowcover to contact-highcover, which based on your data I visualize may have some kind of midpoint maximum, where richness, diversity and dispersion all increase and then decline.

A huge number of different relationships between microbiome response metrics to local and regional algae could be hypothesized. We tried to create a conceptual figure for the most plausible ones, given current knowledge. However, we mentioned that relationships other than those presented in Fig 1 were possible. Additionally, the pattern that we observed was similar to Fig. 1d, just more extreme. We discuss the unimodal relationship between microbiome diversity and the total algal impact score suggested by our data in the discussion and added an additional conceptual figure (Fig. 7) to illustrate it.

3) It also suggests that examining gradients of community structure (possibly island-specific NMDS axes - see my comments for Figure 4?) along these same axes would be useful. Another gradient of community change you could examine would be something like mean bray-curtis distance from a “control” (nocontact-lowcover) - again see my comments for Figure 4.

We already tested the response of coral microbiome composition to algal contact and site-level macroalgal cover, which we believe addresses this comment. However, we have added statistics for that analysis to Fig. 4, and added a panel showing how the response of Bray-Curtis dissimilarity between coral vs. algae samples responds to macroalgal cover and algal contact.

Minor Revisions:

Title: why not change “site-level” to “regional”...this both sounds better and better reflects your model structure (which used site as a random effect)?

The other reviewer does not like the term regional, so we have tried to constrain its usage to only the parts of the introduction and discussion in which we are talking in an abstract, general sense to contrast large vs. small spatial scales.

References: reference 6 and 22 are the same.

We have fixed this issue.

L54-56: I would add that Kelly et al 2014 also showed how benthic cover gradients in algal dominance related to the taxonomic composition of the coral reef microbiome...while not specifically an examination of regional effects on the CORAL microbiome this study did set the stage for benthic cover influencing reef microbiomes orthogonal to biogeochemical and physical parameters.

We have added text to mention Kelly’s study in lines (74).

L117-119: I would also argue that it is not just the population of Mo’orea but also the influence of Tahiti (ie daily traffic)...not sure how best to phrase or quantify but worth pointing out to the uninitiated so it is apparent that Moorea is not an isolated island of 17k people but a satellite of the most populous island in the south pacific.

We have added text to mention Moorea’s proximity to Tahiti.

L175: What is meant in this sentence? What is being “determined”?

We changed the wording to clarify that we measured the DNA concentrations of the cleaned and purified PCR products.

It now reads: Cleaned amplicon library concentrations were measured on a Nanodrop 1000 or a Denovix DS-11 FX+ (Denovix, Wilmington, DE)

L188: You never indicate whether you worked primarily with relative abundances, particularly with corncob and the permanova. This is important.

In the sections describing the PERMANOVA and ordination, we added “calculated from the relative abundance of microbial taxa within each sample” to the sentence, “...we compared microbial community composition of our samples in ordination space using non-metric multidimensional scaling (NMDS), based on the weighted Bray-Curtis dissimilarity matrix...” (PERMANOVA is based on the Bray-Curtis dissimilarity matrix that we calculated based on relative abundances.)

The corncob models (beta-binomial regressions) used raw abundance (read) data and the total number of counts per sample to model relative abundances of taxa. These models use a beta random variable to model binomial probabilities, which allows for overdispersion. Raw abundances are used in conjunction with an overdispersion parameter to account for differences in read depth.) We have edited the corncob section to hopefully clarify that raw read counts are used.

L213: I think you mean dispersion, not beta diversity (your permanovas are also analyzing beta diversity)

We changed this to beta dispersion.

L288: I am skeptical of the utility of the samples with less than several hundred reads, though I could be convinced. One way to explore this is a graphic relating read count to richness...in theory there will be some asymptote above which read count arguably does not appreciably influence richness...I advocate applying a minimum threshold there and excluding from all analyses samples below that number of reads.

We have changed our alpha analyses to use Shannon and Simpson diversity, which are less sensitive to low sampling depth (Chao and Shen 2003). Additionally, we constructed rarefaction curves for our samples, which we have now added to the supplement (Fig. S5). These curves rapidly saturated for all samples, including those with the lowest read depth.

L289: Please provide a supplemental reads/sample distribution was so we have some idea of how much variation and general statistics (mean, median, minimum, maximum). Note that means can't be calculated on non-normal data, so more than likely you need to log₁₀ the reads/sample to get a good mean (or use a geometric mean).

We have added the median to our description in the main text and to Table S3. We have added Fig. S4 showing the distribution of reads per sample.

Means *can* be calculated for non-normal data, although they can be less representative of the central tendency of the data distribution than a median. If we transformed the data, calculated the mean of those transformed values, then back-transformed the mean to the original scale, this would not give us the same mean as calculating the mean of untransformed data. Therefore, we

do not think providing this back-transformed value provides an intuitive summary of the data distribution. We have included medians as an estimate of the central tendency.

L290: In part to solve the problem in L288 it can be useful (and I advise) to apply a threshold of prevalence and maximum relative abundance below which ASVs are excluded from alpha diversity analyses. Please consider, especially if the graph I suggest on L288 makes you feel like you will lose a lot of samples you might get a better graph if you eliminate ASVs that are only in a few samples (say <3) or never have a relative abundance greater than X in more than 1 sample (X might be 0.1%)...this can smooth out the effects of imbalanced sampling effort (reads/sample).

We had no reason to believe that rare ASVs were spurious, therefore removing them would only reduce the accuracy of our estimation of alpha diversity and lower the read depth of our samples. Using Shannon and Simpson diversity as our primary metrics of alpha diversity makes our results relatively robust to imbalanced sampling effort and low read depth (Chao and Shen 2003, Lemos et al. 2013).

L293: Clarify how you “rarefied”. Presumably this was not by true rarefaction (a modeling approach to predict richness at a lower sampling effort == reads/sample) but by random subsampling at a standardized reads/sample effort: give that number and be clear about the approach. You declare 74 reads in the legend for Figure S4...this is very low (See my comment on L288) and it would be better if it were a higher number. I think most microbial ecologists like 10,000, but I am comfortable with a few hundred or ideally 1000. Coral microbiome papers often have low reads because corals are a pain to amplify...

To achieve even sequencing depth for each richness comparison, we randomly subsampled the reads for each sample by the minimum observed read depth in the samples being compared. However, we have removed this from the text now that we are not using richness as a response variable.

Most of our coral samples had higher read depths (median = 31,859), but we had enough that were < 1000 reads that we would not be able to test our hypotheses on the coral samples if we removed these low-read samples. Our rarefaction curves (now added to Fig. S5) suggest that we still were able to adequately capture the species diversity of our coral samples, therefore we chose to keep the low-read samples.

Figure 1: I would label the Y as “microbiome metric” rather than “microbiome response” as this seems like you are saying that the magnitude of the response, rather than the magnitude of a variable (say richness) changes with algal density.

We changed this to “microbiome response variable.”

Figure 2: I don't understand why you would do this: “Significant main effects are not shown if the interaction is significant”. I suspect my recommendation that you put the lmer/adonis2 stats on the figures will eliminate the need for this comment.

We did this because a significant interaction implies that the main effects cannot be interpreted without considering the context of both interacting predictor variables. Therefore, we showed significant interactions on the plot, and only main effects for significant factors not involved in a significant interaction, as recommended by most statisticians. We believe this simplification makes the figures easier to read and interpret.

L382-383: See my comments on Figs S7-S8 about medians on untransformed data.

We address this in our response to comments about Figs. S7-S8.

Figure 5 and 6: I would label the x-axis to clarify the differential test. By this I mean your figure should be interpretable without reading the legend. You state in the legend for Figure 5 “Positive numbers indicate taxa that increased with site-level macroalgal cover, negative numbers indicate taxa that declined with macroalgal cover” so just label one side of the x-axis “increased with macroalgal cover” and the other side “declined with macroalgal cover”. On Figure 6 label one side “increased with algal contact” etc...

We added this to the plot.

Supplement - why is PCR protocol and filtering bioinformatics not in main manuscript? They are quite short, and the filtering bioinformatics are even redundant with the text in L179-181

We were trying to reduce the length of the main manuscript, but we have moved the PCR protocol back to the main text, and now only have the filtering parameters in the supplement.

Supplement - Why in PCR protocol is there “10 s at 78 °C” - unusual, inexplicable, and should be justified.

This was part of the PCR clamps procedure used on the algal samples to block the amplification of sequences from the eukaryotic host (Lundberg et al. 2013. Practical innovations for high-throughput amplicon sequencing). Now that the PCR protocol is back in the main text, this is clearer.

L234-235: Why are you using chi-square tests instead of lmer test or emmeans to evaluate p-values and contrasts? I am familiar with the car package but just haven't used it and don't know why it would apply a chi-squared test. I'd appreciate your advice here...I am just not familiar with the approach.

Wald Chi-squared tests are just another way to estimate p-values for mixed models (see Bolker et al. 2009 reference that we cite in the text). Additionally, emmeans does not support models fit by glmmADMB, which we used for the beta regressions of the beta dispersion data.

L232: I am not fond of this approach because it doesn't really match to testing the hypotheses driving your study. I advocate you structure specific models to test specific hypotheses rather than just whittling the largest models based on stepwise approaches...those are designed for a

different question such as “what suite of variables best describe variation” not “how do variables A, B, C interact to predict Y”...the latter fits your deliberate study design much better.

We were specifically interested in testing for an interaction between local vs. regional algal effects, but we also thought there could be differences in the interaction between islands, which is why we fit models started with a three-way algal contact x macroalgal cover x island interaction, and fit a reduced model containing with algal contact x macroalgal cover and island as a fixed effect when the interaction was not significant. This approach was also a compromise that minimized 1) the reduction of statistical power and 2) the increased risk of Type I errors associated with doing all analyses separately by island. We have edited our text in the methods to explain our hypotheses and this modeling approach more precisely.

Figure S7 and S8 should be presented after transform to normal distribution: you can't put normal quantile box plots on non-normal data, and relative abundances are rarely normal (and they certainly aren't here). Moreover a reader can't really understand these distributions when most of the families hover near zero. Statistically I recommend using the angular transform ($\text{asin}(\sqrt{x})$) but graphically **log10** is much easier to visualize: for visualization with log10 you can convert all zeros to a minimum based on your subsampling/rarefaction level (ie a limit of detection - say 0.01% or something). But if/when you do statistical tests (which I don't think you are doing on these?) it is best to use another transformation to approximate normality such as angular transform. Note that the “median” criteria for inclusion should likewise be calculated on transformed data.

By definition, medians and quantiles (as used by our boxplots) can be used for any data distribution. Medians can also be used as estimates of central tendency for non-normal distributions. We believe that the untransformed data showing relative abundance for specific samples is actually a more direct and intuitive description of the true data distributions, and are appropriate given that we were only summarizing the distributions and did not do any statistical analyses on this data. However, as suggested, we have changed our figures to use a log10 transformation (+ .0001 for zeroes) to visualize relative abundances. To your point about median criteria for inclusion: it is common practice in studies investigating “core” microbial taxa to use prevalence and/or untransformed relative abundances values as criteria for designation as a core taxon (Hernandez-Agreda 2017).

(Hernandez-Agreda 2017. Defining the Core Microbiome in Corals' Microbial Soup. Trends in Microbiology. doi:10.1016/j.tim.2016.11.003.

Figure S8c I recommend “Clade_I” be annotated with what group it actually is (Probably “SAR11 Clade_I” based on figs 5-6).

Changes have been made to Fig S8c to fix this.

Figures 5,6,S7,S8: In general I recommend organizing the order of families on Y axis by Phylum/Class and annotating the order/classes so the reader has some frame of reference (you add the order on Figs 5-6 but not S7-S8). You might also just double-check that “Cyanobiaceae” is indeed the family for what I assume is *Synechococcus*?

We have changed our plots to show class, order, family.

Cyanobiaceae is a family within the order Synechococcales and contained sequences for *Synechococcus*, *Prochlorococcus*, and *Cyanobium* in our coral dataset. Cyanobiaceae is used in at least two curated reference sequence databases, SILVA, which we used, and the Genome Taxonomy Database, GTDB r89. However, within both of these databases, different sequences identified as *Synechococcus* are classified into multiple families, including Synechococcaceae and Cyanobiaceae.

Signed: Craig Nelson (in case you want to reach out for clarification or advice)

Appendix C

Amy A. Briggs
Odum School of Ecology
University of Georgia
140 E. Green St.
Athens, GA 30606
amy.briggs@uga.edu

Editorial staff
Royal Society Open Science

June 1, 2021

Dear Editor(s):

I am resubmitting the original research article, RSOS-210035 “Local vs. site-level effects of algae on coral microbial communities,” by myself and Drs. Anya L. Brown and Craig W. Osenberg for your consideration for publication in *Royal Society Open Science*. Thank you for the opportunity to revise it.

We have made several major changes to the manuscript to address the concerns of the reviewers and the handling editor, Dr. Nicole Hynson. These changes included reporting our alpha diversity results as Hill numbers, adding additional panels to Fig. 4 to help visualize the shift in the similarity of coral vs. macroalgae microbiomes with site-level macroalgal cover, re-running our community composition analyses separately for each island, and adding a table and figures to the supplement to show the read depth distributions and diversity rarefaction curves for our samples. We also added an additional conceptual figure to the discussion to illustrate the relationship between total algal exposure (combining both local and regional effects) and coral microbiome diversity, as suggested by our results. Finally, we re-organized and streamlined parts our results section. Additional minor changes are described in our responses to reviewers. We believe these changes have improved the manuscript, so we are grateful to the reviewers and editor for their constructive and helpful feedback.

Following this letter are the reviewer comments with our responses underneath in blue text. Revisions were made in consultation with all coauthors, and each author has given approval to the final form of this revision. We have no conflicts of interest to declare.

Best regards,

Amy Briggs

Responses to reviewers for manuscript
“Local vs. site-level effects of algae on coral microbial communities”
by

Amy A. Briggs, Anya L. Brown, Craig W. Osenberg

Key: Editor and Reviewer comments in black text. Author responses in blue text.

Associate Editor Comments to Author (Dr Nicole Hynson):

In line with the two expert Reviewer's comments I enjoyed reading this article and feel that it makes a nice contribution to our understanding of some of the factors that govern coral microbiomes. Similar to Reviewer 2, I would suggest the following: rather than reporting observed richness and Shannon diversity, consider reporting the first 3 Hill numbers (of which the first is observed richness, the second is Shannon entropy, and the third inverse Simpsons). Hill numbers have numerous advantages over other diversity indices commonly used for microbial studies. These are outlined in: Chao A, Gotelli NJ, Hsieh TC, Sander EL, Ma KH, Colwell RK, Ellison AM. 2014. Rarefaction and extrapolation with Hill numbers: a framework for sampling and estimation in species diversity studies. *Ecological Monographs* 84: 45–67. and can be easily calculated in the R package *iNext*.

Also, please be sure that you reference the relevant figures in your discussion as well as address the other, mostly minor, comments raised by both Reviewers.

I look forward to reading a revision of this MS.

We thank the editor and reviewers for their detailed and thoughtful comments. As the editor has suggested, we have graphically reported our alpha diversity results as Hill numbers, rather than diversity indices. However, due to statistical characteristics of the data in the Hill number format, we have had to use log-transformed versions of this data for our statistical analyses (which converts Hill number order 1 back to Shannon entropy). We only reported the Hill numbers of order 1 and 2 (Shannon and Simpson diversity), but did not report richness (Hill number order 0), as we believe our sampling depth was too low to accurately estimate richness for a very small subset of samples (which is why we previously subsampled to an even read depth for our richness analyses). Additionally, we used two, rather than three Hill numbers to appease the concerns of Reviewer 2 that multiple alpha diversity results would be correlated. We thank the editor for information about the *iNEXT* package. However, the extrapolated diversity estimates using *iNEXT* did not demonstrably differ from our observed values, even for low-read depth samples, so we used our observed data.

We have addressed the comments of the reviewers by adding two graphs to Fig. 4, adding many additional annotations and statistical output to our graphs, re-doing some statistical analyses by island, and re-organizing our results section. Below we provide specific responses to the reviewers.

Reviewer comments to Author:

Reviewer: 1

Comments to the Author(s)

Briggs et al. use 16S microbial data to test how coral microbiomes shift in response to changes in algal cover (across regional scales) and intimate contact between corals and algae or its absence. The authors present several well articulated hypotheses to explain their results and some nice conceptual figures as well. Ultimately the authors find that many metrics used to study coral microbiomes (alpha diversity, beta diversity) increase with macroalgae community % cover whereas the opposite is observed when corals are in close contact with algae AND algae % cover is increasing. These results suggest that changes in coral microbiomes are both a function of intimate contact with algae, community macroalgae cover, and their interaction. This work, therefore, contributes to our understanding of coral microbiomes and the species-interactions (as well as community traits at varying scales) that lead to shifting microbial community composition.

Overall I found the paper well written, the experiment executed correctly and with appropriate statistical analyses (hats off to you! It is quite impressive). I would rate my comments as minor and seeking clarity overall.

Specifically, I find the terms "local effect and regional effect" to be confusing, as local to me implies a small spatial scale but not immediate proximity (i.e., touching). I don't want to discourage the authors from using the terms they want, but to me describing these terms, showing term fidelity, and re-emphasizing the "intimate contact" or proximity is key here. Similarly, the authors use both "site" and "regional" to describe larger spatial patterns across islands. I think again some term fidelity and using appropriate terms to best describe these effects when discussing them would be helpful. This need for clarity in terms and describing effects is, in my opinion, the most significant issue the manuscript faces, although this is easily addressed.

We agree that "local" and "regional" are relative terms that can refer to different specific scales in different studies. We have tried to be more consistent in using "algal contact" and "site-level macroalgal cover" in our methods, results, and discussion, but we have kept local vs. regional in our introduction and some areas of our discussion when we are discussing these spatial scales in a more general or theoretical sense.

The results section is quite dense, but perhaps not overly so considering the amount of data to unpack. Perhaps some streamlining could be undertaken, but I appreciate if this cannot be done. I also appreciate the author's use of the supplement to provide further information on their study and its findings, which helped provide a lot of valuable information.

We have attempted to streamline the section regarding the responses of specific microbial families in the coral samples to algal contact and macroalgal cover.

In terms of data accessibility, a suggestion: it might be prudent to look into a 3rd party site to host the code/data (such as zenodo or Dryad, which I believe has a deal with RSOS). I see these files in the supplement, but a standalone repo would allow for the code (R project/Rmd/data) to be stored in a succinct directory making it easier for others to access and reproduce analyses.

Now that we have finalized our analyses based on reviewers' comments, we have moved our data and code to Dryad.

Well done and I look forward to seeing this paper published.

Thank you, we appreciate the kind feedback!

Specific comments.

Line 12: spatial scales – for clarity, I think the authors mean “are these effects context dependent based on algal density”, which may not necessarily be spatially explicit but rather determined by a across a range of species-interaction. Perhaps rephrase to emphasize the 2 levels you are focusing on (1) are micorbial interactions varying across spatial scales, (2) and does this relate to macroalgae/turf density. I think you say this quite elegantly in Line 88.

We have a more specific meaning than the reviewer suggests, as density can be evaluated at multiple spatial scales (e.g., algae density in a 1 x 1 m quadrat or algal density in a 100 x 100 m site). In the sentences that the reviewer references, “microbes mediate negative effects of algae on corals when corals are in contact with algae. However, it is unknown whether these effects extend to larger spatial scales, such as at sites with high algal densities,” we think that it is clear that we are contrasting effects of algae on corals at smaller spatial scales (contact) with effects at larger spatial scales, and we also provide specific examples of what those two scales are.

Line 14: colon in stead of comma? (same at Line 90). Could flip to say Moorea and Mangareva, FP (country last—as you do in Line 115).

This would appear to be a matter of style and we would prefer to retain our original usage: “two islands in French Polynesia, Moorea and Mangareva.”

Also, here and elsewhere you have an apostrophe in Moorea (Mo'orea), when it should be an okina (Mo'orea).

We were unaware of this distinction, thank you for pointing this out. We have replaced all apostrophes in Mo'orea with an okina.

Line 20-21: what is “local algae”? density? In Line 21 perhaps rephrase: coral microbiomes are affected by both site-level algae density (in absence of direct contact) and proximity effects between coral and algae. As written the take home isn't punching through.

“Proximity effects” implies evaluation of the effects of algae along a continuous distance scale, which we didn’t do. We are also constrained by the word limit for the abstract. We have therefore retained our original language which is more succinct.

Line 26-28: “Coral reefs” (many different types of ‘reefs’). Citation for the slip-to-slime, as it were?

We added the word “coral” to be more specific, and have added additional citations related to the disturbances threatening coral reefs, and factors leading to algal proliferation.

Line 36/39: hereafter “microbiome?” To orient the reader to the term...

We changed the text to clearly define the term microbiome here, and have tried to use it more throughout the paper.

Line 54: is this to say the size of a site or the distance between sites? I also see the reference to “a local effect” in line 53 with the “local scales” in line 58 to be confusing. There is the local neighborhood (<1000 m²?) and the local neighbor (touching you in this case). Maybe there is a way to disentangle this semantically (i.e. how you discuss the “zone of interaction” in Line 61).
Line 59: citation to clarify this passage?

Old line 54: This is a rough estimate of the area (hence the units, m²) characterized by the sampling design at each site in the cited studies. Area estimates are based on the transect length, the number of transects, and the distance between transects at each site. Distance between sites does not seem to be a good way to characterize a site’s spatial extent, as there can be a large amount of unsampled area between sites (some of which may be open ocean, or not analogous habitat).

A local effect is an effect that occurs over a local spatial scale. Local is defined as close proximity or touching. We have added “(< 1 m²)” to “experimental studies that have demonstrated effects of algae on coral microbiomes all focus on corals that were in close proximity (< 1 m²) or in contact with algae: i.e., a “local” effect.”

Old Lines 58-59: We have changed “...dilution and benthic boundary layers might prevent effects at larger spatial scales.” to, “...effects at larger (*i.e., regional*) spatial scales.” However, this is a hypothetical statement, so we cannot provide a citation to it, as requested.

Line 63: can you again remind us what the distinction is between local and regional SITE scales? You describe this well in Line 344 heading as “local contact vs. site-level effects”. I suppose you are also discussing within site and across a region here too (or is site = region? Seems Line 370 is stating this is the case...)? Some clarification would help.

At the beginning of this paragraph, we link our definition of a regional scale to sites: “Although there are no studies that document effects of algae on corals at larger “regional” scales (e.g., among sites that are each 1,000-10,000 m²).” Our intent is to contrast the absence of studies at

these scales, while there are studies conducted at the local scale, which we mentioned just prior to this sentence: "...corals that were in close proximity or in contact with algae: i.e., a "local" effect..."

Line 68-75: rephrase: (line 68) algae effects on coral microbiomes increase with regional density of algae? In general I find it hard to follow when "algae effect" is stated but what it is affecting is not—ie., "algae effects on coral microbiomes..." (as in Line 73). Specifying the in-contact (<5cm?), local (<1000m²), region (>1000m²?) is important earlier to make this statement in Line 75 clear.

We added ... "on the coral microbiome..." to the next sentence to emphasize what the algae are influencing. We use "local" and "regional" here rather than "in contact" and "site-level" because we are discussing these interactions in the abstract sense, and conceivably, local effects could arise from direct contact with, or close proximity to, algae (e.g., < 1 m²). We are not defining the spatial limits of these effects with this study. Instead, we investigate effects at two contrasting scales, so we want to use terms for the scales to be more general in our introduction. We are, however, more specific in our methods and results.

Line 108: last part of sentence here is not clear.

We have changed the sentence "Contrary to previous hypotheses, we found that algae affected multiple aspects of coral microbiomes at both local and regional scales, and that these different scales modified one another's effects." to "...we found that algae affected multiple aspects of coral microbiomes at both local and regional scales, *and that the effects of algae at one scale modified its effects at the other scale.*"

Line 119: I'd be cautious here, as these effects on islands are very often modified around the island. i.e., more human impact in Cook's Bay with high sewage or groundwater discharge? This would be low-to-absent on south shore...

We have added text "at least some of its reefs" to the statement about differences in anthropogenic influence.

Line 138: missing a ")" after refs. Changes made.

Line 159: use the micro symbol. Changes made.

Line 190: no italics on reference (and again at line 211, 246). Changes made.

Line 198: citation as [44-46]? I'd make sure refs all formatted correctly at final version, such as here and Line 213 "[50,51]"

We have checked our citation formats.

Line 223: “with” perhaps instead of “and” site as a random factor. Changes made.

New paragraph at the start of Line 223 sentence?

We started a new paragraph for the discussion of why we used macroalgal cover as a predictor.

Line 253: “no-local scale”? or what does non-focal mean here?

Non-focal means the scale whose effect we are not testing in this analysis. In these analyses, to evaluate the effect at one scale, we control for the effect at the other (non-focal) scale. We have tried to clarify this in the text.

Line 255: “0.05” Changes made.

Line 309 and elsewhere: there should always be a zero before a decimal place. Please correct throughout. Changes made.

Line 355: the “like it did” and “narrowly missed the cutoff” seems a bit too informal. Clarify the effect you are referencing and rephrase non-significant results as a trend?

Because we have re-run the PERMANOVA separately by island, our results and the associated text have changed.

Line 383: be consistent with or without spacing before “%”. We have inserted a space between a number and the % symbol throughout manuscript.

Line 410: comma after e.g. Changes made.

Line 412: algae density at regional scales? Coral microbiomes responding to this or something else (a different reef substrate)?

We have changed the words “due to” to “with” to emphasize this is a correlation. Additionally, we mention in the discussion that other site-level factors could be driving the association between site-level macroalgal cover and changes in the coral microbiome.

Line 441-442: I don't see the phrasing here as making the point more clear. Can this be clarified? Perhaps even simply putting “switch from negative (in contact with algae) to positive (no alga contact)” would suffice.

The suggested rephrasing is not accurate, so we have retained the original wording, but replaced “regional” with “site-level” to address the reviewer's other concerns.

Line 443: I see this is strongly correlative, not that this is bad, but high macroalgae cover may be one of many things at a site/region that are affecting microbial communities. I'd perhaps caution

the “cause and effect” interpretation—as you eloquently put in the following paragraph and in the conclusion passage.

We have changed the word “were” to “could be” to clarify that the hypothetical nature of our interpretation in this sentence.

Line 499: or the abiotic conditions that favor microbial growth?

We have added the text, “change abiotic conditions and/or...” to include changes other than increases in algal by-products.

Line 517: no longer clades, say species/genera
Changed to species

Line 537: by site-level effect you really mean algal abundance, yes? Maybe saying the term instead of the effect would help in clarifying these passages.

Here, “site-level effect of macroalgae” means the effect of macroalgal cover averaged across the site-scale. We have changed the text to reflect this.

Line 555: local and site-level algae? You mean intimate contact or macroalgae abundance?

Yes. We changed the text to “algal contact and site-level macroalgal cover” to be more explicit.

Line 560: is it better to describe these groups as “putative” beneficial symbionts (as you do in Line 505)? I’m uncertain whether these groups are well linked to function/health the way this sentence suggests.

We added the word “putative.”

All figures in main text and supplement—Moorea needs okina as well. Changes made.

Fig. 1. I really like this figure! Thank you.

Fig 2. Can you spell out interaction or define it in the legend?

We have added definitions of our abbreviation for macroalgal cover (macro. cover) to the caption and changed alg. contact to algal contact.

Reviewer: 2

Comments to the Author(s)

Review attached because the system crashed on me.

Review of Amy Briggs, Anya Brown, Craig Osenberg for Proc. Open Science. “Local vs. site-level effects of algae on coral microbial communities”

Briggs et al present a stellar study of interacting effects of local and regional algal influence on coral microbiomes.

Note to the editor: this paper is one I would nominate for a cover story or other highlight, as this is a very strong and widely interesting manuscript.

The conceptualization of this study is excellent and couched within a strong literature framework. The experimental design is solid and the statistical analyses are well done. I have some issues with a few aspects of the data analysis and presentation which I consider major revisions, but I expect that the authors will be able to make these changes and improve the manuscript to a level appropriate to the audience of this journal. Overall I think this is an excellent piece of work and I believe that the authors are underselling this.

We thank the reviewer for his kind words and thoughtful review.

Please note: I did not carefully review the discussion, because I believe it should be restructured and may need to be rewritten based off of some of my recommendations. I will review it during revision.

Major Points:

I advocate that Shannon Diversity be replaced or at least complemented with an evenness metric (Shannon Evenness or Pielou's J). Shannon Diversity and Richness are highly correlated in this study (which is common when richness gradients dominate the variation in Shannon Diversity Index) and are redundant, while evenness will at least be orthogonal.

The editor and both reviewers had contrasting suggestions about how to handle these analyses. We have attempted to address their collective comments in spirit (if not exactly as proposed), given the disparate advice. We believe Shannon diversity (Hill number order 1) and Simpson diversity (Hill number order 2) are better estimates of alpha diversity than unrarefied richness (Hill number order 0) in our study, due to the low read depth for a portion of our samples. Shannon Diversity incorporates evenness and can accurately estimate alpha diversity even at low sampling intensity. It is also robust to zeroes and skewed relative abundances of taxa (Chao and Shen 2003, Lemos et al. 2013).

Since evenness is correlated with both Shannon and Simpson diversity, we have decided not to separately evaluate an evenness metric. Describing the change in the effective number of taxa for each Hill number allows us to discuss how dominant taxa contribute to diversity, which will address some of Reviewer 2's request that we explicitly consider evenness. A decline in the Hill number with increasing orders of q indicates that the community is uneven.

Chao, A., Chiu, C.- H., & Jost, L. (2014). Unifying species diversity, phylogenetic diversity, functional diversity, and related similarity and differentiation measures through Hill numbers. *Annual Review of Ecology Evolution and Systematics*, 45, 297–324.

Chao, A.; Shen, T. 2003. Nonparametric estimation of Shannon's index of diversity when there are unseen species in sample. *Environ. Ecol. Stat.*, 10, 429-433.

Lemos LN, Fulthorpe RR, Triplett EW, Roesch LFW. 2011 Rethinking microbial diversity analysis in the high throughput sequencing era. *J. Microbiol. Methods* 86, 42–51. (doi:10.1016/j.mimet.2011.03.014)

Restructure so that all the models are presented together. It is reasonable to present univariate and multivariate graphics separately, but the model structures (lmer and adonis2) are nearly identical (although adonis2 does not allow the random effect of site) and can be presented to dovetail alpha and beta diversity. In this way the results and discussion can be separated mainly by model structure (there are 2 different frameworks, see my comment below) and thus by hypotheses being tested. In other words why not combine Table S3 and S4+S5 with Tables S6 and S7, respectively?

Along these same lines, when presenting models and results it is valuable to be explicit about the hypotheses being tested. This will allow the reader to better understand the modeling framework and goals.

We have reorganized our results section into a section comparing microbiome responses among substrate types and islands, and a section evaluating the effects of local vs. site-level algae on microbiome responses. We have also tried to be more specific about our hypotheses in the methods section when we describe our modeling approach.

Putting p values (and for adonis2/PERMANOVA also R^2) on graphs for fixed effects in multivariate models will help a great deal. The supplement should only be a reference, not critical to interpreting the graphics. If you want to leave things like DF and F statistics in the supplement in tables that is fine, but the core statistics should be obvious on the graphics. I would expect to see appropriate lmer results on Figure 2, S4, S5 and adonis2 results on Figures 3 and 4 and S6.

We appreciate that it is easier to have statistical results indicated on the graphs, at least when the figure and the statistics are simple. We have done that in figures in which we think the statistics will not detract from the readability of the graphs (e.g., Fig. 2, 3) However, adding too much text to a plot detracts from the ability to read it, and we were unable to add all the requested statistics to some of our figures using a legible font size (e.g., adding PERMANOVA results to Fig. 4).

Since the statistical results are provided in the main text, and we indicate in figure captions where to look at in the supplement to find statistical summaries, we believe we have done due diligence to help the reader find the results. We also do not feel that it is necessary to add results to supplemental graphs when the table summarizing those results is on the next page.

Given that you analyzed different species of coral, different species of macroalgae, and most likely different assemblages of turfing algae and different water masses, I would argue that you “know” that “island” will be a significant effect and you simply need to evaluate whether there is an island*cover interaction term. Since there is a universally significant interaction term, you can report these but then frame your subsequent analysis differently: Table S3 and Table S6 effectively argue that Tables S4-S5 and S7-S8 should be run separately for each island, with Table S4 and the “coral parts” of Tables S7-S8 being run to include the interaction of contact and cover.

The specific identities of taxa contributing to diversity metrics at each island did not affect the diversity calculations (e.g., the corals at each island could have completely different taxa in their microbiomes but still have similar richness or beta dispersion). Additionally, we did not necessarily know that diversity would be demonstrably different by island, so we thought it was a predictor worth testing. When we tested models containing an island x macroalgal cover interaction, this interaction was *not* significant for any of our diversity metrics in Fig 2 (corals) or for our algae and water samples. Therefore, we did not do separate tests by island for the alpha diversity and beta dispersion data, which would have reduced the power of using our whole dataset to estimate our parameters, and inflated the probability of Type I errors by increasing the number of statistical tests that we performed.

The cornucopia analyses were separated by island because these models dealt with the relative abundances of specific microbial taxa, which were not always found in both coral species. However, since community composition deals with specific microbial taxa, we have now run separate analyses for each island to test the effects of local algal contact and macroalgal cover.

Along these lines, it is worth noting that Figure 4 implies a PERMANOVA (adonis2) model that doesn't exist as far as I can tell from Table S7, but it should given your hypotheses in Figure S1: Figure 4 implies a model like “For each Island BCdistance~cover*contact” where you report R² and p on each panel for all three terms. I advocate you run this model and provide those statistics to support the visualization provided in Figure 4.

We did perform a PERMANOVA on the coral samples testing if Bray Curtis dissimilarity of samples was different with site-level macroalgal cover and algal contact. These results were summarized in the text, and we indicated in the figure caption the supplemental table that summarized the PERMANOVA results. To add the full PERMANOVA results to the plot, the text would need to be too small to be readable. However, we have added the p-values. We also added two panels to Fig. 4 that show the position of coral and macroalgae samples along the primary axis separating the coral and algae samples, NMDS1, vs. site-level macroalgal cover. These panels show the pattern we described in the text, and we have added the results for a regression fit to the data for each island testing the effects of macroalgal cover and algal contact on NMDS1.

Overall, it is my feeling that the manuscript will read much smoother if you consider the above and follow this structure: I think the figures are fine as is (with some tweaks as noted and just add the stats as I suggested):

1) alpha diversity (richness and evenness by lmer) and beta diversity (dispersion lmer and distance adonis2) ~ island*cover : results shown in Figures 3 and S4 with statistics on the figures for each of the three fixed effects (p for lmer, p and R2 for adonis2).

2) Only Corals, separated by island, alpha diversity (richness and evenness by lmer) and beta diversity (dispersion lmer and distance adonis2) ~ cover*contact : results shown in Figures 2 and 4 with stats on the figures for each of the three fixed effects (p for lmer, p and R2 for adonis2) - Note that crucially this demands that the ordinations in Figure 4 be done separately for each island, which I currently think they are not (which is inappropriate regardless).

We did ordinations separately by island in Fig 4. We used Shannon and Simpson diversity rather than richness and evenness (justification in our response to the first major point in the review). We did not run separate statistical tests by island for diversity (justification in our response to the earlier comment, "Given that you analyzed different species of coral,...").

Figure 4 and L359-367: This is an important test of some of your hypotheses (particularly Figure S1) and the discussion should be more quantitative and less qualitative. NMDS are a visualization tool, not statistics, so I would endeavor to support these statements with statistical data before asking the reader to see the pattern in an NMDS. You can run pairwise.adonis which might allow you to test if the distance between treatments nocontact and algae is smaller or larger than the distance between turfcontact or macroalgaecontact and algae at your three different macroalgae cover levels (which may need to be condensed?).

Pairwise tests would allow us to test if our groups (either coral local contact groups or coral local contact groups vs. macroalgae) are significantly different and if they respond to macroalgae cover differently, however, we still need to use a graphical display to determine if the dissimilarities are larger for one group vs. another.

I also advocate considering a regression approach, where the bray-curtis distance between coral and algae (y) is related to %cover separately for each of the three contact types separately for each island. You can do this by decomposing your distance matrix and annotating all pairwise bray-curtis distances between these categories on each island as one of the three types (nocontact-algae, turfcontact-algae, macrocontact-algae).

PERMANOVA for the dissimilarities between coral vs. algae samples was not possible, because PERMANOVA requires a square distance matrix, meaning coral vs. coral and algae vs. algae dissimilarities would also need to be included. Raw Bray-Curtis dissimilarities were not used in a regression because they were not independent (1000s of combinations of coral vs. algae samples for each island). The best (albeit, statistically imperfect) approach we could take to test the dissimilarity between coral vs. macroalgae microbiomes was to fit linear regression models containing algal contact, site-level macroalgal cover, and their interaction, to the NMDS1 response for each coral and macroalgae sample at each island. However, NMDS ordination

distances are not independent, so results should be interpreted cautiously. We have added this last approach (and associated caveat) to our results and Fig. 4.

Finally, for your thought and consideration, I find Figure 2 compelling and want you to think about some aspects and ways to bring this concept to the forefront, perhaps even with a new conceptual framework for how algal impacts work (and even how local vs. regional stressors work). I accept that you may not agree with or be able to find these patterns, but I do think you should consider them because your data appear stronger and more interesting than you give them credit for.

We describe a new conceptual framework that could explain the patterns in Fig 2 in the discussion (with the colonization/differential growth tradeoffs). We believe it is most appropriate to discuss this framework in there, rather than the introduction, as we do not know of any evidence published prior to our study to suggest a greater-than-antagonistic interaction between local and regional algal effects. We also did not measure some of the variables (like microbial density) that would be necessary to test the new mechanistic framework that we proposed in our discussion (e.g., effects of algae on microbial densities influencing colonization rates).

1) I think it would be worth testing if the Turf vs. Macroalgal contrast is significant (doesn't look like it) and then you can report that and on the figure just contrast coral with algaecontact. Will make a much clearer story with just two regression lines instead of three.

There are some differences in the responses of beta dispersion and microbial community composition to contact with macroalgae vs. turf algae (e.g., slope differences in response of beta dispersion to macroalgal cover for the two contact groups). Additionally, we have a priori reasons to believe the different types of algae could have different effects. Therefore, we chose not to aggregate the data.

2) It seems that you may have a pattern that isn't in figure 1, namely local-regional compensation, where high enough regional effects (high enough algal diversity) overcome a lack of local effects and position the response variables at the "start" of local impacts (ie direct algal contact with minimal regional effects). In this sense, you might envision an additive continuous "algal impact" score, from nocontact-lowcover to nocontact-highcover to contact-lowcover to contact-highcover, which based on your data I visualize may have some kind of midpoint maximum, where richness, diversity and dispersion all increase and then decline.

A huge number of different relationships between microbiome response metrics to local and regional algae could be hypothesized. We tried to create a conceptual figure for the most plausible ones, given current knowledge. However, we mentioned that relationships other than those presented in Fig 1 were possible. Additionally, the pattern that we observed was similar to Fig. 1d, just more extreme. We discuss the unimodal relationship between microbiome diversity and the total algal impact score suggested by our data in the discussion and added an additional conceptual figure (Fig. 7) to illustrate it.

3) It also suggests that examining gradients of community structure (possibly island-specific NMDS axes - see my comments for Figure 4?) along these same axes would be useful. Another gradient of community change you could examine would be something like mean bray-curtis distance from a “control” (nocontact-lowcover) - again see my comments for Figure 4.

We already tested the response of coral microbiome composition to algal contact and site-level macroalgal cover, which we believe addresses this comment. However, we have added statistics for that analysis to Fig. 4, and added a panel showing how the response of Bray-Curtis dissimilarity between coral vs. algae samples responds to macroalgal cover and algal contact.

Minor Revisions:

Title: why not change “site-level” to “regional”...this both sounds better and better reflects your model structure (which used site as a random effect)?

The other reviewer does not like the term regional, so we have tried to constrain its usage to only the parts of the introduction and discussion in which we are talking in an abstract, general sense to contrast large vs. small spatial scales.

References: reference 6 and 22 are the same.

We have fixed this issue.

L54-56: I would add that Kelly et al 2014 also showed how benthic cover gradients in algal dominance related to the taxonomic composition of the coral reef microbiome...while not specifically an examination of regional effects on the CORAL microbiome this study did set the stage for benthic cover influencing reef microbiomes orthogonal to biogeochemical and physical parameters.

We have added text to mention Kelly’s study in lines (74).

L117-119: I would also argue that it is not just the population of Mo’orea but also the influence of Tahiti (ie daily traffic)...not sure how best to phrase or quantify but worth pointing out to the uninitiated so it is apparent that Moorea is not an isolated island of 17k people but a satellite of the most populous island in the south pacific.

We have added text to mention Moorea’s proximity to Tahiti.

L175: What is meant in this sentence? What is being “determined”?

We changed the wording to clarify that we measured the DNA concentrations of the cleaned and purified PCR products.

It now reads: Cleaned amplicon library concentrations were measured on a Nanodrop 1000 or a Denovix DS-11 FX+ (Denovix, Wilmington, DE)

L188: You never indicate whether you worked primarily with relative abundances, particularly with corncob and the permanova. This is important.

In the sections describing the PERMANOVA and ordination, we added “calculated from the relative abundance of microbial taxa within each sample” to the sentence, “...we compared microbial community composition of our samples in ordination space using non-metric multidimensional scaling (NMDS), based on the weighted Bray-Curtis dissimilarity matrix...” (PERMANOVA is based on the Bray-Curtis dissimilarity matrix that we calculated based on relative abundances.)

The corncob models (beta-binomial regressions) used raw abundance (read) data and the total number of counts per sample to model relative abundances of taxa. These models use a beta random variable to model binomial probabilities, which allows for overdispersion. Raw abundances are used in conjunction with an overdispersion parameter to account for differences in read depth.) We have edited the corncob section to hopefully clarify that raw read counts are used.

L213: I think you mean dispersion, not beta diversity (your permanovas are also analyzing beta diversity)

We changed this to beta dispersion.

L288: I am skeptical of the utility of the samples with less than several hundred reads, though I could be convinced. One way to explore this is a graphic relating read count to richness...in theory there will be some asymptote above which read count arguably does not appreciably influence richness...I advocate applying a minimum threshold there and excluding from all analyses samples below that number of reads.

We have changed our alpha analyses to use Shannon and Simpson diversity, which are less sensitive to low sampling depth (Chao and Shen 2003). Additionally, we constructed rarefaction curves for our samples, which we have now added to the supplement (Fig. S5). These curves rapidly saturated for all samples, including those with the lowest read depth.

L289: Please provide a supplemental reads/sample distribution was so we have some idea of how much variation and general statistics (mean, median, minimum, maximum). Note that means can't be calculated on non-normal data, so more than likely you need to log₁₀ the reads/sample to get a good mean (or use a geometric mean).

We have added the median to our description in the main text and to Table S3. We have added Fig. S4 showing the distribution of reads per sample.

Means *can* be calculated for non-normal data, although they can be less representative of the central tendency of the data distribution than a median. If we transformed the data, calculated the mean of those transformed values, then back-transformed the mean to the original scale, this would not give us the same mean as calculating the mean of untransformed data. Therefore, we

do not think providing this back-transformed value provides an intuitive summary of the data distribution. We have included medians as an estimate of the central tendency.

L290: In part to solve the problem in L288 it can be useful (and I advise) to apply a threshold of prevalence and maximum relative abundance below which ASVs are excluded from alpha diversity analyses. Please consider, especially if the graph I suggest on L288 makes you feel like you will lose a lot of samples you might get a better graph if you eliminate ASVs that are only in a few samples (say <3) or never have a relative abundance greater than X in more than 1 sample (X might be 0.1%)...this can smooth out the effects of imbalanced sampling effort (reads/sample).

We had no reason to believe that rare ASVs were spurious, therefore removing them would only reduce the accuracy of our estimation of alpha diversity and lower the read depth of our samples. Using Shannon and Simpson diversity as our primary metrics of alpha diversity makes our results relatively robust to imbalanced sampling effort and low read depth (Chao and Shen 2003, Lemos et al. 2013).

L293: Clarify how you “rarefied”. Presumably this was not by true rarefaction (a modeling approach to predict richness at a lower sampling effort == reads/sample) but by random subsampling at a standardized reads/sample effort: give that number and be clear about the approach. You declare 74 reads in the legend for Figure S4...this is very low (See my comment on L288) and it would be better if it were a higher number. I think most microbial ecologists like 10,000, but I am comfortable with a few hundred or ideally 1000. Coral microbiome papers often have low reads because corals are a pain to amplify...

To achieve even sequencing depth for each richness comparison, we randomly subsampled the reads for each sample by the minimum observed read depth in the samples being compared. However, we have removed this from the text now that we are not using richness as a response variable.

Most of our coral samples had higher read depths (median = 31,859), but we had enough that were < 1000 reads that we would not be able to test our hypotheses on the coral samples if we removed these low-read samples. Our rarefaction curves (now added to Fig. S5) suggest that we still were able to adequately capture the species diversity of our coral samples, therefore we chose to keep the low-read samples.

Figure 1: I would label the Y as “microbiome metric” rather than “microbiome response” as this seems like you are saying that the magnitude of the response, rather than the magnitude of a variable (say richness) changes with algal density.

We changed this to “microbiome response variable.”

Figure 2: I don't understand why you would do this: “Significant main effects are not shown if the interaction is significant”. I suspect my recommendation that you put the lmer/adonis2 stats on the figures will eliminate the need for this comment.

We did this because a significant interaction implies that the main effects cannot be interpreted without considering the context of both interacting predictor variables. Therefore, we showed significant interactions on the plot, and only main effects for significant factors not involved in a significant interaction, as recommended by most statisticians. We believe this simplification makes the figures easier to read and interpret.

L382-383: See my comments on Figs S7-S8 about medians on untransformed data.

We address this in our response to comments about Figs. S7-S8.

Figure 5 and 6: I would label the x-axis to clarify the differential test. By this I mean your figure should be interpretable without reading the legend. You state in the legend for Figure 5 “Positive numbers indicate taxa that increased with site-level macroalgal cover, negative numbers indicate taxa that declined with macroalgal cover” so just label one side of the x-axis “increased with macroalgal cover” and the other side “declined with macroalgal cover”. On Figure 6 label one side “increased with algal contact” etc...

We added this to the plot.

Supplement - why is PCR protocol and filtering bioinformatics not in main manuscript? They are quite short, and the filtering bioinformatics are even redundant with the text in L179-181

We were trying to reduce the length of the main manuscript, but we have moved the PCR protocol back to the main text, and now only have the filtering parameters in the supplement.

Supplement - Why in PCR protocol is there “10 s at 78 °C” - unusual, inexplicable, and should be justified.

This was part of the PCR clamps procedure used on the algal samples to block the amplification of sequences from the eukaryotic host (Lundberg et al. 2013. Practical innovations for high-throughput amplicon sequencing). Now that the PCR protocol is back in the main text, this is clearer.

L234-235: Why are you using chi-square tests instead of lmer test or emmeans to evaluate p-values and contrasts? I am familiar with the car package but just haven't used it and don't know why it would apply a chi-squared test. I'd appreciate your advice here...I am just not familiar with the approach.

Wald Chi-squared tests are just another way to estimate p-values for mixed models (see Bolker et al. 2009 reference that we cite in the text). Additionally, emmeans does not support models fit by glmmADMB, which we used for the beta regressions of the beta dispersion data.

L232: I am not fond of this approach because it doesn't really match to testing the hypotheses driving your study. I advocate you structure specific models to test specific hypotheses rather than just whittling the largest models based on stepwise approaches...those are designed for a

different question such as “what suite of variables best describe variation” not “how do variables A, B, C interact to predict Y”...the latter fits your deliberate study design much better.

We were specifically interested in testing for an interaction between local vs. regional algal effects, but we also thought there could be differences in the interaction between islands, which is why we fit models started with a three-way algal contact x macroalgal cover x island interaction, and fit a reduced model containing with algal contact x macroalgal cover and island as a fixed effect when the interaction was not significant. This approach was also a compromise that minimized 1) the reduction of statistical power and 2) the increased risk of Type I errors associated with doing all analyses separately by island. We have edited our text in the methods to explain our hypotheses and this modeling approach more precisely.

Figure S7 and S8 should be presented after transform to normal distribution: you can't put normal quantile box plots on non-normal data, and relative abundances are rarely normal (and they certainly aren't here). Moreover a reader can't really understand these distributions when most of the families hover near zero. Statistically I recommend using the angular transform ($\text{asin}(\sqrt{x})$) but graphically **log10** is much easier to visualize: for visualization with log10 you can convert all zeros to a minimum based on your subsampling/rarefaction level (ie a limit of detection - say 0.01% or something). But if/when you do statistical tests (which I don't think you are doing on these?) it is best to use another transformation to approximate normality such as angular transform. Note that the “median” criteria for inclusion should likewise be calculated on transformed data.

By definition, medians and quantiles (as used by our boxplots) can be used for any data distribution. Medians can also be used as estimates of central tendency for non-normal distributions. We believe that the untransformed data showing relative abundance for specific samples is actually a more direct and intuitive description of the true data distributions, and are appropriate given that we were only summarizing the distributions and did not do any statistical analyses on this data. However, as suggested, we have changed our figures to use a log10 transformation (+ .0001 for zeroes) to visualize relative abundances. To your point about median criteria for inclusion: it is common practice in studies investigating “core” microbial taxa to use prevalence and/or untransformed relative abundances values as criteria for designation as a core taxon (Hernandez-Agreda 2017).

(Hernandez-Agreda 2017. Defining the Core Microbiome in Corals' Microbial Soup. Trends in Microbiology. doi:10.1016/j.tim.2016.11.003.

Figure S8c I recommend “Clade_I” be annotated with what group it actually is (Probably “SAR11 Clade_I” based on figs 5-6).

Changes have been made to Fig S8c to fix this.

Figures 5,6,S7,S8: In general I recommend organizing the order of families on Y axis by Phylum/Class and annotating the order/classes so the reader has some frame of reference (you add the order on Figs 5-6 but not S7-S8). You might also just double-check that “Cyanobiaceae” is indeed the family for what I assume is *Synechococcus*?

We have changed our plots to show class, order, family.

Cyanobiaceae is a family within the order Synechococcales and contained sequences for *Synechococcus*, *Prochlorococcus*, and *Cyanobium* in our coral dataset. Cyanobiaceae is used in at least two curated reference sequence databases, SILVA, which we used, and the Genome Taxonomy Database, GTDB r89. However, within both of these databases, different sequences identified as *Synechococcus* are classified into multiple families, including Synechococcaceae and Cyanobiaceae.

Signed: Craig Nelson (in case you want to reach out for clarification or advice)

Appendix D

**Responses to reviewers (2nd revision) for manuscript
“Local vs. site-level effects of algae on coral microbial communities”**
by

Amy A. Briggs, Anya L. Brown, Craig W. Osenberg

Key: Editor and Reviewer comments in black text. Author responses in blue text.

Associate Editor Comments to Author (Dr Nicole Hynson):

The authors have done a sufficient job of revising their MS based on the Reviewer's comments and defending their rationale for not making some of the suggested changes. Reviewer 1 has a few minor comments to be addressed before publication and Reviewer 2 points out that the proper diacritical mark to honor the Tahitian written language would be an apostrophe in Mo'orea. I request that the authors make these minor changes.

Thank you. We have made all the changes requested by the reviewers.

Reviewer: 1

Comments to the Author(s)

The authors have addressed all my comments and concerns and have made substantial improvements to the manuscript which have helped the both the structure and ease of interpretation (especially in the methods). My comments are quite minor and I apologize to the authors for taking so long to return my review. However, I wanted to give proper attention to the revision and the comments of reviewer 2 (who is much more of an expert than I am!). In short, I am happy to recommend the MS for publication.

PS - Excellent work on the conceptual figures. I think they really help communicate the findings and hypotheses. Thank you.

My comments are on the clean version of the MS not the tracked changed version FYI.

As a small note it appears the “C” in °C in PCR protocol (see Line 31) is a different font than rest of paper. Thank you pointing this out—the font change was not apparent in the Word document, but did show up in the pdf. We changed the “°C” format so that it should now show up correctly.

Line 38 page 9: what is “1:30s”? Is this 90s? 1.5min? We changed this to 1 min 30 s

Line 10 page 10: “v.” for version We added a “.” to “v”

Line 4-22 page 12: this seems to be information for the results section. Perhaps you moved this

around due to a request from the 2 reviewers, but it seems it would be best suited to go with results of PCA. If you used this as a way to test how your main effects should be structured/included in models then I see that as perhaps reasonable to include here. It is not clear to me how PCA was used (results are NMDS) so perhaps a little context as to why you used PCA would help. You do a nice job of describing the results of the PCA but less on the “why” which would be important here in the methods.

We used PCA to further justify our selection of macroalgal cover as a predictor variable in our statistical models of microbiome diversity, composition, etc. (This analysis supported our methodological approach, but we did not consider it a major research question we were seeking to address, so we placed it in the methods.) However, we see how this could be confusing, so we have moved text from the supplement explaining how we performed the PCA to the Methods in the main text, and moved the results of the PCA to the Results section.

Line 38: page 17: there is a “_” before the sentence starts. We have removed this underline.

Line 45: underline the bold subheading like above sections? (and sans colon) We have added an underline to the bold subheadings and replace the colon in “Responses of specific microbial families:” with a period.

Reviewer: 2

Comments to the Author(s)

The revised manuscript is excellent, the discussion well written, and the revisions appropriate (including the excellent explanations about the decision making process in addressing the sometimes disparate recommendations of the reviewers and editor). Both reviews were extensive and I think improved the manuscript significantly. I thought that the discussion on Figure 2 was well considered, and appreciate that the authors considered my suggestions thoughtfully and were clear when and why they disagreed.

I respect the disagreement with the authors on presenting means of non-normal data and whether this is valid. I don't disagree that calculating the mean of non-normal data can be useful, but I do disagree that comparing means or drawing quantile distributions like box-whisker diagrams is inappropriate unless the data are transformed to meet the assumptions of those approaches. Nonetheless, the authors are clearly well-versed in statistics and I applaud their determination to stick to their approaches and clearly articulate their reasoning, so I think this is all fine and good.

Thank you. We appreciate all the thoughtful, specific, and helpful feedback to improve our manuscript.

Fun note:

Although Reviewer 1 suggested "Also, here and elsewhere you have an apostrophe in Moorea (Mo'orea), when it should be an okina (Mo‘orea)." and you responded (appropriately for the request) as follows "We were unaware of this distinction, thank you for pointing this out. We have replaced all apostrophes in Mo‘orea with an okina." ...Reviewer 1 is actually sort of wrong,

and you were sort of right. The grammar differs among polynesian languages, though all have some kind of consonant that represents a hardening of the palate (aka glottal stop), and in Tahitian the cognate of the hawaiian 'okina ('eta) is written as an apostrophe (whereas in Hawaiian the 'okina is written using the ASCII symbol for an "open quote")...so you are correct either way. Most Tahitian writing uses an apostrophe, so Mo'orea is most common.

Thank you for this clarification. We have revised our manuscript to use the apostrophe, rather than the okina.